# High-Dimensional Safe Exploration via Optimistic Local Latent Safe Optimization

## Abstract

Optimization over high-dimensional input space is inherently difficult, especially when safety needs to be maintained during sampling. Current safe exploration algorithms ensure safety by conservatively expanding the safe region, which leads to inefficiency in large input settings. Existing high-dimensional constrained optimization methods also neglect safety in the search process. In this paper, we propose Optimistic Local Latent Safe Optimization (OLLSO), which is capable of handling high-dimensional problems under probabilistic safety satisfaction. We first use distance-preserved autoencoder to transform the original input space into a low-dimensional continuous latent space. An optimistic local safe strategy is then applied over the latent space to efficiently optimize the utility function. Theoretically, we prove the probabilistic safety guarantee from the latent space to the original space. OLLSO outperforms representative high-dimensional constrained optimization algorithms in simulation experiments. We also show its real application in clinical experiments for safe and efficient online optimization of a neuromodulation therapy.

## 1 Introduction

Bayesian optimization is often used for black-box optimization problems where evaluations are expensive, such as hyperparameter tuning and experimental design (Snoek et al., 2012; Hernández-Lobato et al., 2017). This framework typically uses Gaussian Processes (GP) (Rasmussen, 2003) as a surrogate model to estimate function distributions, and optimize an acquisition function to balance exploration and exploitation (Frazier, 2018; Shahriari et al., 2015). Safety-critical systems are common in real-world applications. For example, in robot control, we need to stay away from unsafe states to avoid the potential damage to the expensive equipment. In clinical therapy design, we must avoid the therapies that would potentially hurt the patient. These scenarios correspond to the problem of *safe exploration*, where we need to sequentially optimize an unknown utility function while satisfying some unknown safety constraints. Most existing safe exploration methods discriminate safe regions with estimated function lower confidence bound to ensure safety with high probability (Sui et al., 2015; 2018; Turchetta et al., 2019; Baumann et al., 2021; Sukhija et al., 2022). Such conservative or pessimistic strategies might be inefficient in high-dimensional and large-scale input settings, which are common in real-life scenarios. Further more, the search space may contain both discrete and continuous variables, which is difficult to utilize common-used GP kernel function to model input similarities.

A motivating application of our work is the control of human movement via neuromodulation, where therapists need to sequentially select parameters of a high-dimensional electrode array implanted in the human body. Typically the search region consists of 16 discrete dimensions representing contact configurations (each contact can be set to be positive, negative or unused) and one continuous dimension representing stimulation intensity. Under this high-dimensional hybrid input setting, efficiently optimizing the task function while maintaining safety remains a challenge for existing safe optimization algorithms. Despite the considerable attention given to use Bayesian optimization methods for solving the high-dimensional constrained optimization problem, they often fail to incorporate safety considerations into the optimization procedure (Griffiths & Hernández-Lobato, 2020; Notin et al., 2021; Eriksson & Poloczek, 2021). To our best knowledge, there are no methods that can guarantee safety, or probabilistic safety during the high-dimensional optimization.

In this paper, we propose Optimistic Local Latent Safe Optimization (OLLSO) to address safety over high-dimensional sequential optimization problems. To deal with high-dimensional hybrid inputs, OLLSO uses a regularized autoencoder to map the original structured input space into a continuous latent space while preserving distances. Over the latent space, the algorithm optimizes the objective function using an *optimistic local safe* strategy, discriminating safe regions of a local search space with estimated upper confidence bounds of the safety function. We derive the theoretical probabilistic safety guarantee of OLLSO from the latent space to the original space. We applied the algorithm to two high-dimensional safety critical problems. It achieved more efficient optimization performance and safer sampling procedure compared to existing high-dimensional constrained Bayesian optimization algorithms. We deployed OLLSO in real clinical experiments, and successfully optimized the lower limb muscle control of a paraplegic patient.

## 2 RELATED WORK

### 2.1 SAFE BAYESIAN OPTIMIZATION

The sequential decision-making problem with safety constraints has been extensively studied, varied by the definition of safety. To achieve full safety during exploratory sampling, algorithms have been proposed with theoretical guarantee in near-optimality and safety with high probability (Sui et al., 2015; Turchetta et al., 2019). These methods have been applied in safe parameter tuning of quadrotor (Berkenkamp et al., 2021) and Swiss free electron laser (Kirschner et al., 2019). In robot control, two recent papers achieved global safe parameter optimization by learning backup policies during exploration (Baumann et al., 2021; Sukhija et al., 2022). These safe optimization algorithms conservatively estimate and expand the safe region, leading to inefficient optimization performance.

In contrast to a zero-tolerance approach to unsafe actions, an alternative approach allows for limited constraint violations within a predefined budget, trading safety for more efficient optimization. Incorporating an additional penalty term is a commonly-used method to restrict constraint violations(Zhou & Ji, 2022; Lu & Paulson, 2022; Guo et al., 2023). The effectiveness of this method is closely tied to the selection of penalty parameters, including the penalty weight and dual update step size. A recent work uses upper confidence bound to optimistically estimate the safe region, enjoying global optimal guarantee as unconstrained methods (Xu et al., 2022; 2023).

Another extreme case, called constrained Bayesian optimization, aims only to find the best feasible solution, neglecting the safety during the optimization process (Gardner et al., 2014; Gelbart et al., 2014; Hernández-Lobato et al., 2016; Marco et al., 2020; 2021). Constrained Expected Improvement (cEI) is a popular constrained BO algorithm that introduces feasibility constraints to acquisition function formulation (Schonlau et al., 1998; Gelbart et al., 2014).

All of the aforementioned methods fall in the framework of Bayesian optimization (BO), which is typically limited to low-dimensional problems with continuous input space (Frazier, 2018; Shahriari et al., 2015). LineBO (Kirschner et al., 2019) demonstrates success in optimizing problems with dimension up to 40, but not compatible with discrete inputs. In high-dimensional settings, pessimistic safe algorithms such as Sui et al. (2015; 2018); Turchetta et al. (2019) might even struggle to expand the safe region due to sparse discretization of the input space.

### 2.2 HIGH-DIMENSIONAL BAYESIAN OPTIMIZATION

Over the last few years, Bayesian optimization has been used in high-dimensional problems (Turner et al., 2021; Binois & Wycoff, 2022). In this section, we focus on dimension reduction-based and local BO methods. Further discussion about other lines of work can be found in the Appendix D.

A large body of literature leverages dimension reduction to apply BO over a low-dimensional subspace. Several works use variable selection to identify and optimize important dimensions during optimization(Chen et al., 2012; Zhang et al., 2019; Song et al., 2022). Another popular approach for reducing the search space is random linear embedding, which as been proven to contain the optimal solution with a certain probably relative to the objective function's effective dimension. This method has gained significant attention and support in the literature, as evidenced in the works (Wang et al., 2016; Nayebi et al., 2019; Li et al., 2018; Letham et al., 2020; Papenmeier et al., 2022).

Additionally, deep autoencoder models, such as variational autoencoder (VAE) are powerful tools for learning continuous representations from high-dimensional structured data (Kingma & Welling, 2013). Many works also use autoencoders to learn a non-linear mapping between the original space and the latent space (Gómez-Bombarelli et al., 2018; Moriconi et al., 2020; Tripp et al., 2020; Deshwal & Doppa, 2021; Grosnit et al., 2021; Siivola et al., 2021; Notin et al., 2021). Such dimension reduction methods have also been successfully applied to high dimensional constrained optimization problems, which are often used to relieve the invalid input issue when projecting points back to the original space. Works such as Griffiths & Hernández-Lobato (2020) apply cEI over the latent space to sample valid molecule sequences with a higher probability.

Another line of work utilizes local search to optimize over the high-dimensional input space, achieving better empirical performance than global BO methods(Eriksson et al., 2019; Müller et al., 2021; Nguyen et al., 2022; Wu et al., 2023). Several works use the trust region method to address the over-exploration issue in high-dimensional optimization(Eriksson et al., 2019; Wang et al., 2020). This local optimization strategy is also able to be deployed in latent space optimization (Maus et al., 2022) and high-dimensional constrained optimization problem (Eriksson & Poloczek, 2021).

Besides BO methods, evolutionary algorithms such as CMA-ES are competitive to solve high-dimensional problems (Hansen, 2006). They can also handle constrained optimization problems after simple modifications (Kramer, 2010; Arnold & Hansen, 2012). Although many works attempt to solve high-dimensional constrained optimization problems, to the best of our knowledge, there lacks work that addresses safety in high-dimensional sequential optimization.

## 3 PROBLEM FORMULATION

### 3.1 BLACK-BOX OPTIMIZATION WITH PROBABILISTIC SAFETY CONSTRAINTS

We aim to optimize an unknown objective function $f : \mathcal{X} \to \mathbb{R}$ by sequentially sampling points $\boldsymbol{x}_1, \ldots, \boldsymbol{x}_n \in \mathcal{X}$. We can also get observations of safety measurement from other unknown functions $g_1, \ldots, g_m : \mathcal{X} \to \mathbb{R}$. We define a point $\boldsymbol{x}$ is safe when $\forall i \in [1, m], g_i(\boldsymbol{x}) > h_i$, where $h_1, \ldots, h_m$ are pre-defined safety thresholds. For clearer explanation we set the number of safety functions to 1, and denote safety function as $g(\boldsymbol{x})$ and threshold as $h$. In this paper, we aim to improve sample efficiency by slightly relaxing the safety constraint to allow a small number of unsafe decisions. In this sense, we introduce a probabilistic version of the safety constraint, which requires that each sample point is safe with a probability above predefined threshold $\alpha$. We can formally write our optimization problem as follows:

$$\max_{\boldsymbol{x}_t \in \mathcal{X}} f(\boldsymbol{x}_t) \quad \text{subject to } \Pr(g(\boldsymbol{x}_t) \geq h) \geq \alpha, \forall t \geq 1, \tag{1}$$

where the $\alpha$ is usually smaller than $0.5$, which indicates the worst case of safety violation we can tolerate. Furthermore, the input space $\mathcal{X} \in \mathbb{R}^D$ may be high-dimensional in real-world applications, and may consist of both discrete and continuous variables. Commonly-used kernel functions struggle to well represent similarity between inputs.

### 3.2 LATENT SPACE BAYESIAN OPTIMIZATION

To deal with high-dimensional data, one way is to learn a mapping between original input space $\mathcal{X}$ and a low-dimensional continuous latent space $\mathcal{Z} \in \mathbb{R}^d$. If we can find a low-dimensional latent space in which distances in the original high-dimensional space could preserve, the safety boundary can be properly estimated within this latent space. Latent space optimization sequentially samples points $\boldsymbol{z}_1, \ldots, \boldsymbol{z} \in \mathcal{Z}$, and evaluates the function values by decoding the sampled points: $w_f(\boldsymbol{z}) = f(\mathcal{D}(\boldsymbol{z}))$ and $w_g(\boldsymbol{z}) = g(\mathcal{D}(\boldsymbol{z}))$ for objective and safety functions respectively. To optimize over the latent space, we make the following assumptions that are commonly used in the field of Bayesian optimization, taking $w_g$ as an example.

**Assumption 1.** *Functions over the latent space $\mathcal{Z}$ have bounded norm in the associated Reproducing Kernel Hilbert Space (RKHS). Let $k : \mathbb{R}^d \times \mathbb{R}^d \to \mathbb{R}$ be symmetric, positive-semidefinite kernel functions, $w_g \in \mathcal{H}_k, \|w_g\| < B$.*

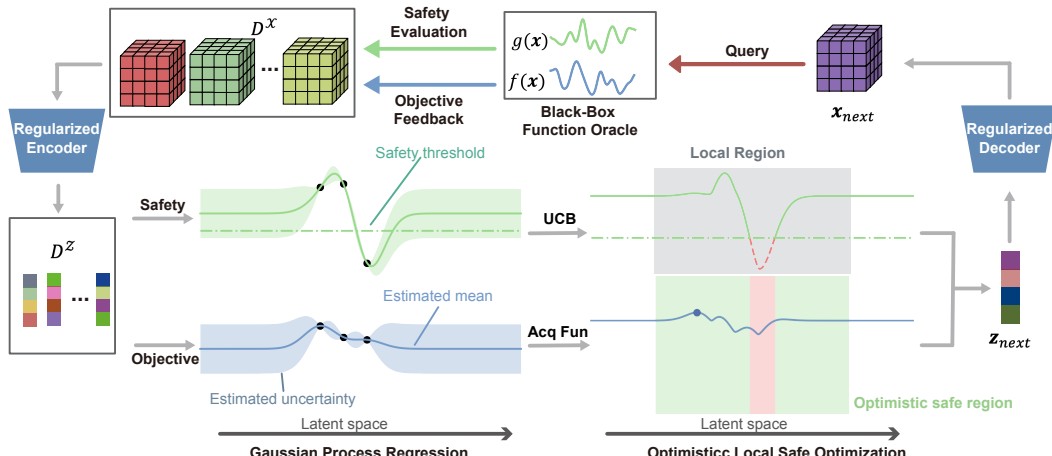

Figure 1: Workflow of OLLSO. We use a regularized autoencoder to enable optimization over the latent space and evaluation over the original space. Optimistic local safe optimization is used over the latent space to efficiently optimize the objective function while guaranteeing probabilistic safety.

**Assumption 2.** *Functions observations are perturbed by i.i.d. Gaussian noise:* $y^g(\boldsymbol{z}_t) = w_g(\boldsymbol{z}_t) + n_t$, *where* $n_t \sim \mathcal{N}(0, \sigma^2)$

Under the above assumptions, we are able to use a Gaussian process as the surrogate model to learn the unknown functions. For samples at points $A_T = [\boldsymbol{z}_1...\boldsymbol{z}_t]^T$, we have noise-perturbed observations $\boldsymbol{y}_T = [y_1^g...y_t^g]^T$. The posterior over $w_g$ is also Gaussian with mean $\mu_t(\boldsymbol{z})$, covariance $k_t(\boldsymbol{z}, \boldsymbol{z}')$ and variance $\sigma_t^2(\boldsymbol{z}, \boldsymbol{z}')$:

$$\mu_t(\boldsymbol{z}) = \boldsymbol{k}_t(\boldsymbol{z})^T (\boldsymbol{K}_t + \sigma^2 \boldsymbol{I})^{-1} \boldsymbol{y}_t$$
$$k_t(\boldsymbol{z}, \boldsymbol{z}') = k(\boldsymbol{z}, \boldsymbol{z}') - \boldsymbol{k}_t(\boldsymbol{z})^T (\boldsymbol{K}_t + \sigma^2 \boldsymbol{I})^{-1} \mathbf{k}_t(\boldsymbol{z}') \tag{2}$$
$$\sigma_t^2(\boldsymbol{z}) = k_t(\boldsymbol{z}, \boldsymbol{z}),$$

where $\boldsymbol{k}_t(\boldsymbol{z}) = [k(\boldsymbol{z}_1, \boldsymbol{z}), ..., k(\boldsymbol{z}_t, \boldsymbol{z})]$ is the covariance between $\boldsymbol{z}$ and sampled points, $\boldsymbol{K}_t$ is the covariance of sampled positions: $[k(\boldsymbol{z}, \boldsymbol{z}')]_{\boldsymbol{z}, \boldsymbol{z}' \in A_t}$. Similarly we can use GP to derive posterior of $w_f$ under same assumptions. Using the posterior of GP, we can define the confidence interval of point $\boldsymbol{z}$ as $C_t(\boldsymbol{z}) := [\mu_t(\boldsymbol{z}) \pm \beta \sigma_t(\boldsymbol{z})]$, where $\beta$ is a scalar which can be properly set to contain $f(\boldsymbol{z})$ with high probability (Srinivas et al., 2009; Chowdhury & Gopalan, 2017). We define $u_t(\boldsymbol{z}) := \max C_t(\boldsymbol{z})$ as the upper confidence bound (UCB) of the function estimation.

## 4 OPTIMISTIC LOCAL LATENT SAFE OPTIMIZATION

### 4.1 ALGORITHM OVERVIEW

We introduce OLLSO, an innovative algorithm designed for probabilistic safety while optimizing in a high-dimensional space. The main structure of this algorithm is shown in Figure 1 and detailed in Algorithm 1. Prior to the optimization loop, an Isometrically Regularized VAE (IRVAE) is trained with unlabeled data $D_u^{\mathcal{X}}$, which can be obtained or synthesized in large quantities. This training is enhanced by an additional regularization loss, emphasizing scaled isometry within the latent space, as discussed by Yonghyeon et al. (2021). The implication of scaled isometry is that the mapping within the latent space preserves angles and distances up to a certain scale factor. Consequently, we expect the GP estimation disparities between the latent and original spaces to be minimal. This ensures that points considered safe by GP in the latent space have a high probability of being safe in the original space. Our experimental results highlight that the IRVAE provides superior GP estimation error reduction when compared to the standard VAE (Kingma & Welling, 2013).

OLLSO begins with training an regularized autoencoder (Line 1) and projecting data features into the latent space (Line 4). With the latent features and function observations in hand, the algorithm estimates the posterior of the objective and safety functions using separate Gaussian processes (Line

5). A local search region, informed by historical data which encompasses sampled observations and the complete sample trajectory, is then defined (Line 6). Utilizing the GP posterior of the safety function, OLLSO confidently delineates a safe space within the local region, guided by the upper confidence bound of the safety function (Line 7). The acquisition function is then optimized over this safe space, with the recommended latent point subsequently projected back to its original input domain using the decoder (Lines 8-9). The history data is updated by assessing the new inputs (Lines 10-12).

Following the standard local Bayesian optimization algorithms, a trust region method is employed to dynamically pinpoint local search regions (Eriksson et al., 2019; Wang et al., 2020; Eriksson & Poloczek, 2021). Specifically, the current safest position is designated as the epicenter of the search to create a safe region with side length $l$. A sampling round is considered "successful" if it finds a better reward while maintaining comprehensive safety. Conversely, it is labeled a "failure" if any unsafe points are found or if there is no discernible improvement. The side length is adjusted—increased for successes and decreased for failures—upon reaching a preset threshold. Unlike conventional local BO methods that discard all data and restart when the side length reaches its minimum, our approach resets $l$ to its initial length, ensuring a different, safer trajectory sampling than the initial instance.

---

**Algorithm 1** Optimistic Local Latent Safe Optimization (OLLSO)

**Input** Sample set $\mathcal{X}$, GP priors $GP^f, GP^g$, safety threshold $h$, acquisition function $A$, unlabelled dataset $D_u^{\mathcal{X}}$, initial dataset $D_0^{\mathcal{X}}$, local region $L_0$
1: Train encoder $\mathcal{E}$ and decoder $\mathcal{D}$ using $D_u^{\mathcal{X}}$
2: Sample trajectory $\zeta_0 \leftarrow \emptyset$
3: **for** $t = 1$ to $\ldots$ **do**
4: $\quad D_{t-1}^{\mathcal{Z}} \leftarrow \mathcal{E}(D_{t-1}^{\mathcal{X}})$
5: $\quad$ Update $GP^f, GP^g$ using $D_{t-1}^{\mathcal{Z}}$
6: $\quad$ Update local region $L_t$ using $D_{t-1}^{\mathcal{X}}$ and $\zeta_{t-1}$
7: $\quad S_t \leftarrow \{z' \in L_t \mid u_t(z) \geq h\}$
8: $\quad z_t \leftarrow \mathrm{argmax}_{z \in S_t}(A(z))$
9: $\quad x_t \leftarrow \mathcal{D}(z_t)$
10: $\quad y_t^f \leftarrow f(x_t) + n_t$
11: $\quad y_t^g \leftarrow g(x_t) + n_t$
12: $\quad$ Update $D_t^{\mathcal{X}}$ and $\zeta_t$ using $x_t, y_t^f, y_t^g$
13: **end for**

---

With respect to the acquisition function $A$, Thompson sampling (TS) was selected (Kandasamy et al., 2018) due to its compatibility with the discrete nature of our safety estimation and search space, and its innate ability for batch optimization by sampling the GP posterior—an appropriate choice for high-dimensional tasks that support parallel evaluations.

## 4.2 Comparison with Previous Work

The significant difference between OLLSO and CONFIG proposed by Xu et al. (2023) is that we constrain the search space within the identified local region, which is crucial for high-dimensional optimization. In terms of the objective optimization, global BO algorithms usually perform worse than local BO algorithms on high-dimensional tasks due to the over-exploration problem (Eriksson et al., 2019). The same issue also affects safety exploration of CONFIG. When there are too many points with high upper confidence bounds due to large posterior uncertainty, CONFIG would identify almost everywhere far from sampled points as safe and degenerate to an unconstrained algorithm. Maintaining an adaptive local region contributes to more efficient optimization and safer search.

Compared to SCBO (Eriksson & Poloczek, 2021), which is a trust region-based BO method for high-dimensional constrained optimization problems, OLLSO considers safety during the optimization procedure by selecting points that are probabilistically safe, and adapts the trust region in a different way. Unlike cEI (Schonlau et al., 1998; Gardner et al., 2014) which directly combines constraints with the acquisition function, OLLSO disentangles the constraints and objective, and optimization the original acquisition function over identified safe region.

## 5 Theoretical Analysis

The safety identification of OLLSO depends on the confidence interval estimate from safety GP. The scalar $\beta_t$ controls the tightness of the confidence bound. Here we also derive the choice of $\beta_t$ to ensure probabilistic safety over the latent space.

**Proposition 1.** *Let Assumptions 1 and 2 hold for the latent safety function $w_g$, and set $\beta_t$ satisfying $\Phi(\beta_t) \leq 1 - \alpha$. Then:*

$$Pr(w_g(\boldsymbol{z}) \geq \mu_{t-1}(\boldsymbol{z}) + \beta_t \sigma_{t-1}(\boldsymbol{z})) \geq \alpha, \forall \boldsymbol{z} \in \mathcal{Z}, \forall t \geq 1, \tag{3}$$

*where $\Phi(\cdot)$ is the cumulative distribution function of the standard normal distribution $\mathcal{N} \sim (0, 1)$.*

The proof uses the property of the Gaussian CDF. The resulting $\beta_t$ does not depend on the maximum information gain, which scales exponentially with the function dimension((Sui et al., 2015)). When $\alpha \to 0.5, \beta_t \to 0$. In practice, choosing a small $\beta_t$ usually satisfies the safety requirement.

When applying safe optimization over the latent space, a natural question is whether safety can be guaranteed in the original space. Here we show that, under certain assumptions, the probabilistic safety guarantee can be extended to the original space. To enable GP-based safety estimation, we need to make assumptions about the regularity of the original space. Like other dimension-reduction based-BO methods, we assume functions to be considered have low-dimensional effective variables (Wang et al., 2016; Nayebi et al., 2019; Papenmeier et al., 2022).

**Assumption 3.** *There exists a mapping $U \in \mathbb{R}^{D \times d_e}$ with orthonormal columns, s.t. $g(\boldsymbol{x}) = g(\boldsymbol{z}U^T)$, where $\boldsymbol{z} = U\boldsymbol{x}$, and $U^T$ is the the inverse mapping of $U$.*

We also want the latent space mapping to approximately preserve the norms of all vectors in the effective subspace spanned by the columns $U$. Here we assume that the latent space mapping is an $\varepsilon$-subspace mapping (Sarlos, 2006; Cohen et al., 2015):

**Definition 1.** *$\Pi$ is an $\varepsilon$-subspace embedding for $U \in \mathbb{R}^{D \times d_e}, U^T U = I$, if $\|(\Pi U)^T (\Pi U) - I\| \leq \varepsilon$. This is equivalent to $\forall r \in \mathbb{R}^{d_e}, (1 - \varepsilon)\|r\|_2^2 \leq \|\Pi r\|_2^2 \leq (1 + \varepsilon)\|r\|_2^2$.*

The definition indicates that $\varepsilon$-embedding is able to preserve the distance between latent and original space up to some constant. This assumption on latent space mapping can be satisfied by linear mappings, such as principal component analysis when the number of principal components is larger than the effective dimension. In the Experiments section, we empirically show that the latent spaces of IRVAE approximately meet the $\varepsilon$-subspace embedding assumption. Under the above assumptions about the distance preserving ability of latent space mapping, we can properly choose $\beta_t$ that satisfies the probability safety guarantee, under the noise-free setting.

**Theorem 1.** *Let assumptions 1 and 3 hold for safety function g, and the latent space mapping $\Pi$ is a $\varepsilon$-subspace embedding for effective subspace basis $U$. At every step t, given candidate set $\boldsymbol{z} \in \mathcal{C}^{\mathcal{Z}}$ and set $\beta_t$ satisfying $\Phi(\beta_t + (5l(\varepsilon)\|X^-\boldsymbol{g}\| + 2\beta_t\sqrt{3l(\varepsilon)})\widehat{\|\boldsymbol{x}\|}/\widehat{\sigma}_{t-1}) \leq 1 - \alpha$, then*

$$Pr(g(\boldsymbol{x}) \geq \mu_{t-1}(\boldsymbol{x}) + \beta_t \sigma_{t-1}(\boldsymbol{x})) \geq \alpha, \boldsymbol{x} \in \mathcal{C}^{\mathcal{X}} \tag{4}$$

*where $X^-$ is the Moore-Penrose pseudoinverse of $X$, and $l(\varepsilon) >= |k(\boldsymbol{x}, \boldsymbol{x}') - k(\boldsymbol{z}, \boldsymbol{z}')|_{\boldsymbol{x} = \Pi^{-1}\boldsymbol{z}}$ is the upper bound of kernel function difference, $\mathcal{C}^{\mathcal{X}} = \{\boldsymbol{x} = \Pi^{-1}\boldsymbol{z}\}_{\boldsymbol{z} \in \mathcal{C}^{\mathcal{Z}}}, \widehat{\|\boldsymbol{x}\|} = \max_{\boldsymbol{x} \in \mathcal{C}^{\mathcal{X}}} \|\boldsymbol{x}\|), \widehat{\sigma}_{t-1} = \min_{\boldsymbol{z} \in \mathcal{C}^{\mathcal{Z}}} \sigma_{t-1}(\boldsymbol{z})$.*

The proof uses the GP estimation difference bound between original and latent space. Using the mean and variance bounds of GP, we bound the difference of UCB between latent and the original space. Then we derive the choice of $\beta$ to satisfy probabilistic safety requirement over the original space according to Gaussian CDF. As $\varepsilon \to 0, \Pi$ becomes a isometric mapping, where choosing the $\beta_t$ same as in Proposition 1 fulfills the safety requirement.

# 6 EXPERIMENTS

## 6.1 EXPERIMENTAL SETUP

We compared OLLSO against the following competitive constrained optimization baselines: SCBO (Eriksson & Poloczek, 2021), CONFIG (Xu et al., 2023), two versions of constrained EI (via blocking the objective function of infeasible points (denoted as cEI, Schonlau et al. (1998)), and via combining acquisition function with feasible probability (denoted as cEI-Prob, Gardner et al. (2014)) and CMA-ES (Hansen, 2006). We also run SafeOpt (Sui et al., 2015) on the problems, but it failed to expand the safe region on all benchmarks we used. We run all the baselines over both original space (suffixed with "O") and latent space (suffixed with "L") on all simulation tasks.

We present the performance of our algorithm in two high-dimensional optimization problems in the main paper: human musculoskeletal system control and spinal cord stimulation, which are safety-critical during the optimization process. We also run other experiments such as constrained handwriting digital generation task (see Appendix C.1). Besides simulation experiments, we applied OLLSO to a real clinical therapy optimization and achieved good safety and efficiency.

We evaluate the performance of the algorithm according to three metrics: best feasible objective function value (Objective), safe decision ratio of all samples (Safe %), and cumulative safety violation (Violation). The plots show the means with one standard deviation.

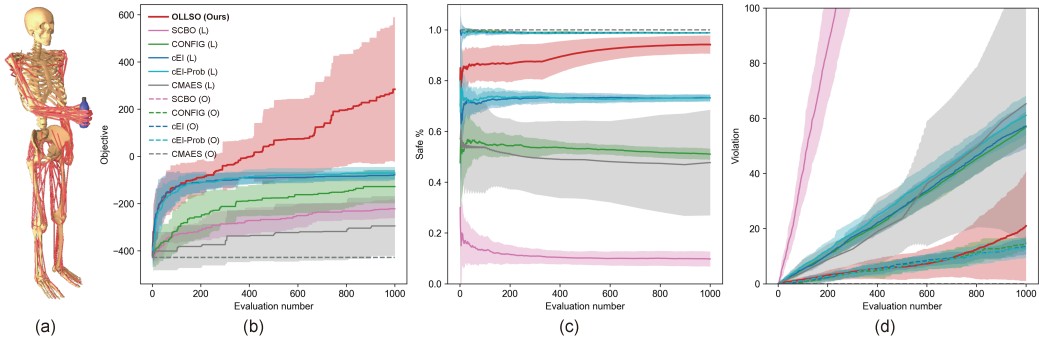

Figure 2: Optimization for the control of a musculoskeletal model. (a) A musculoskeletal model, where the task is actuating hand muscles to hold the bottle vertically and steadily. (b) Best feasible objective function value. (c) Safety decision ratio. (d) Cumulative safety violations. Algorithm performance averaged over 20 independent runs.

## 6.2 OPTIMIZATION FOR THE CONTROL OF A MUSCULOSKELETAL MODEL

In the musculoskeletal control task, one need to optimize the activities of hand related muscles to hold a bottle in vertical position(Figure 2 (a)). Compared to joint-driven problems, controlling muscle-driven models is more challenging because the number of muscles is much larger than the number of joints. As in Mania et al. (2018), we formulate the original reinforcement learning (RL) problem as a sampling problem, where the algorithms need to optimize a linear policy: $\pi \in \mathbb{R}^{|a| \times |o|}$, where $|a| = 55$ and $|o| = 65$ are the dimensions of the action space and the observation space, respectively. The policy to be optimized has $D = 3355$ parameters, which is a very high-dimensional task. We set the objective function as the accumulated reward from the environment, and the safety function as the landing speed of the bottle after it slips from the hand. We collected muscle activation to train a regularized autoencoder to build the muscle synergies of performing the task, reducing action dimension from 55 to 5 (3355 to 325 for policy dimension). While the search space is significantly reduced, the remaining optimization problem is still high-dimensional.

We show the optimization result in Figure 2 (b)-(d). All original baselines fail to make improvement when searching over the extremely high-dimensional input space (curves overlapped in Figure 2 (b)). Even in the latent space, no baseline algorithms can get a positive reward. They also make more unsafe selections and cumulative safety violations than OLLSO. Our proposed method significantly outperforms all latent baselines over three metrics, achieving efficient safe optimization over high-dimensional latent space.

## 6.3 CLINICAL NEURAL STIMULATION

We validate our proposed methods through simulation and clinical experiments of spinal cord stimulation therapy. As in Figure 3 (a), a 32-contact electrode array is implanted outside the patient's spinal dura mater. The electrical stimulation delivered by the electrode array could induce patient's muscle activities thus we could control the lower limb movements of the patient by setting different parameters spatially and temporally. Typically we manipulate half of the contacts to separately control left and right lower limbs, including 16-contact configuration (discrete) and one intensity parameter (continuous). Our optimization goal is the selectivity index of target muscles, computed from 12 groups of muscles (see Appendix B.4). Better selectivity index indicates better control over the target muscle group and less influence over non-target muscles.

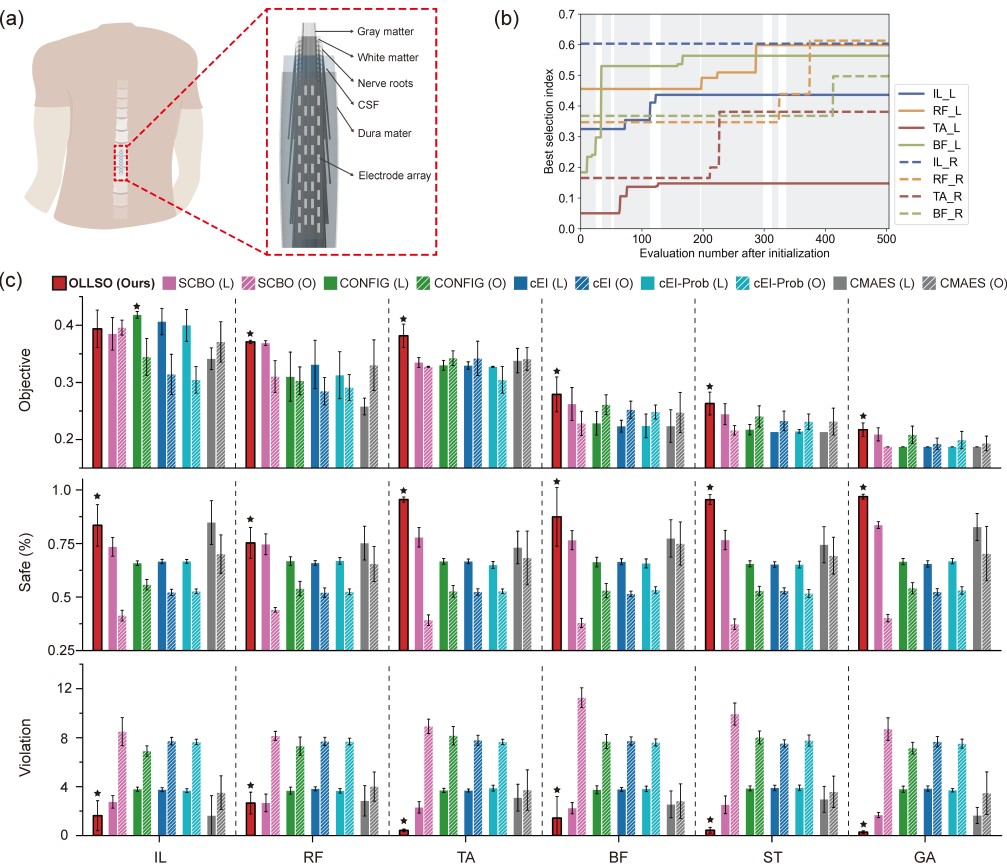

Figure 3: Experiment of spinal cord stimulation. (a) Illustration of the spinal cord model. The model was developed based on anatomical statistics, which contains multiple tissues of the spinal cord and the electrode array. (b) Optimization of the muscle selectivity index in clinical experiment. Shaded areas indicate configurations recommended by OLLSO. IL, RF, TA, BF, ST, GA are different group of target muscle on the left lower limb. Selectivity indices of target muscles increased over trials. shaded area indicates parameter was recommended by OLLSO. (c) Algorithm performance on 6 SCS simulation tasks, averaged over 10 independent runs. The best-performing algorithm was starred on the top of bars.

To enrich the physical meaning of this discrete representation, we transformed 17d vectors into a 2d electric field images with $52 \times 14$ pixels using simplified computation. Then we train an IRVAE with a 16d latent space to embed and reconstruct the electric field map. We also restrict the number of cathode and anode to only evaluate reasonable configurations according to clinical prior.

### 6.3.1 SIMULATION OVER HUMAN SPINAL CORD MODEL

In simulation, we use a human spinal cord model as a function oracle, which is capable of computing the evoked electric field given certain stimulation parameter, and inferring the lower limb muscle activation (Figure 3 (a)). We use the maximum induced muscle activation as the safety function to avoid hurting the patient during optimization. The total sample budget is 1000 including 200 initial random sample parameters.

We show the simulation results in Figure 3 (c). While the dimension of latent space is similar to the original space, latent algorithms execute less unsafe decisions by optimizing on the continuous manifold. OLLSO finds the highest muscle selectivity in 5 out of 6 tasks and causes significantly less cumulative safety violations than all baselines. Our proposed method also maintains the highest the safety selection ratio in all tasks and is the only algorithm that can execute over 95% safety samples in TA and GA task.

### 6.3.2 CLINICAL EXPERIMENTS

We further applied OLLSO to optimize stimulation parameters of a paraplegic patient with the same electrode array implanted. The objectives of our optimization were the stimulation selectivity for 8 target muscles of the lower limb, which could be calculated from Electromyography (EMG). We defined a threshold where safe parameters did not induce pain or large scale lower limb movement. The clinical experiments were approved by the IRB of the hospital.

A total of 636 trials were conducted with the patient over 1 months. In each trial, one parameter was configured on the stimulator, and 8 spikes of stimulation were delivered to the patient. We used the EMG to compute the selectivity index and queried safety scores from the patient and the therapists. The feedback would be added to the data set and the algorithm would recommended the next parameter after each trial was done. We reused previous history data when optimizing a new task.

As shown in Figure 3 (b), we observed selectivity improvement of 7 from 8 target muscles compared to the baseline of initial data (left IL: 0.112, left RF: 0.143, left TA: 0.097, left BF: 0.380, right IL: 0.00, right RF: 0.266, right TA: 0.216, right BF: 0.141). During the whole experimental procedure, only three configurations recommended by OLLSO were rated as unsafe stimulation, which evoked large lower limb movements but no physical damage or pain. The clinical results indicate OLLSO could recommended more selective parameters for different targets while ensuring safety.

### 6.4 DISTANCE PRESERVING OVER LATENT SPACE

In the implementation of OLLSO, we propose to learn a mapping with distance-preserving property. Here we compare the distance-preserving method IRVAE against standard VAE in musculoskeletal model control task. From Figure 4 (a)-(d) we observe IRVAE reduces GP estimation error over latent space in both safety and objective function modeling. Figure 4 (e) shows that this distance preserving properties significantly contributes to OLLSO's optimization performance, where the algorithm using standard VAE fails to make improvement due to large estimation error of objective GP.

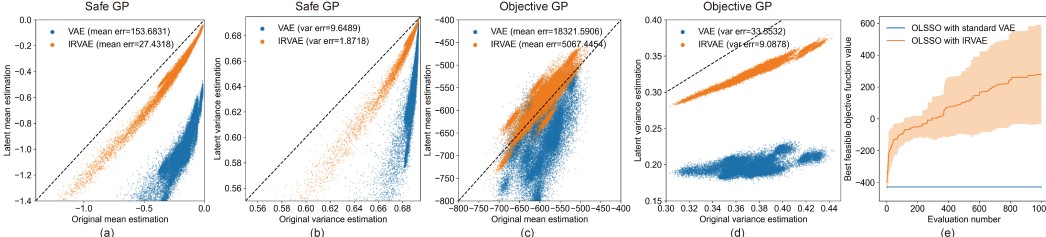

Figure 4: Distance preserving of IRVAE and standard VAE on musculoskeletal model control task. Better scatters are closer to the dashed line in the figure, indicating identical GP estimation between original and latent space. Errors indicate the L2 difference of the safety function estimations between original GP and latent GP.(a)-(b) GP estimation of safety function. (c) GP estimation of safety function. (e) Optimization performance of OLLSO when using different VAEs.

## 7 CONCLUSION

We develop the Optimistic Local Latent Safe Optimization method for safe optimization over high-dimensional spaces. OLLSO uses a regularized autoencoder to map the original structured input space into a continuous latent space while preserving distances, which allows high-dimensional hybrid inputs. An optimistic local safe strategy is used in the latent space to optimize the objective function and expand the safe region. We provide the theoretical probabilistic safety guarantee of OLLSO. We applied the algorithm to safety critical problems. It achieved more efficient optimization and safer sampling compared to state-of-the-art high-dimensional constrained Bayesian optimization algorithms. OLLSO also successfully optimized the lower limb muscle control of a paraplegic patient in real clinical experiments.

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

# A    THEORETICAL ANALYSIS

## A.1    PROOF OF PROPOSITION 1

*Proof.* Fix $t \geq 1$ and $\boldsymbol{z} \in \mathcal{Z}$. Conditioned on $\boldsymbol{y}_{t-1} = (y_1^g, \ldots, y_{t-1}^g)$, $\{\boldsymbol{z}_1, \ldots, \boldsymbol{z}_{t-1}\}$ are deterministic, and $w_g(\boldsymbol{z}) \sim \mathcal{N}(\mu_{t-1}(\boldsymbol{z}), \sigma_{t-1}(\boldsymbol{z})))$. Now if $r \sim \mathcal{N}(0, 1)$, then

$$\Pr(r > c) = 1 - \Phi(c) \tag{5}$$

Therefore, by applying $r = (w_g(\boldsymbol{z}) - \mu_{t-1}(\boldsymbol{z}))/\sigma_{t-1}(\boldsymbol{z})$ and $c = \beta_t$, the statement holds.    $\square$

## A.2    PROOF OF THEOREM 1

We first introduce results from Nayebi et al. (2019) which bounds the GP estimation difference and kernel function difference between the original space and latent space under the noise-free setting.

**Lemma 1.** *(Theorem 2 and Corollary 8 in (Nayebi et al., 2019)) Consider a Gaussian process with kernel function $k$ that acts directly in the unknown active subspace of dimension $d_e$ with mean and variance functions $\mu(\cdot), \sigma^2(\cdot)$. Let $\tilde{\mu}(\cdot), \tilde{\sigma}^2(\cdot)$ be their approximations using an $\varepsilon$-subspace embedding for the active subspace. Given $X = (\boldsymbol{x}_1, \ldots, \boldsymbol{x}_t)$ and $\boldsymbol{g} = (g(\boldsymbol{x}_1), \ldots, g(\boldsymbol{x}_t))$ we have for every $\boldsymbol{x} \in \mathcal{X}$ and $\boldsymbol{z} = \Pi\boldsymbol{x} \in \mathcal{Z}$*

*1. $|\mu(\boldsymbol{x}) - \tilde{\mu}(\boldsymbol{z})| \leq 5l(\varepsilon)\|x\|\|X^-\boldsymbol{g}\|$*

*2. $|\sigma^2(\boldsymbol{x}) - \tilde{\sigma}^2(\boldsymbol{z})| \leq 12l(\varepsilon)\|\boldsymbol{x}\|^2$,*

*where $X^-$ is the Moore-Penrose pseudoinverse of $X$, and $l(\varepsilon) >= |k(\boldsymbol{x}, \boldsymbol{x}') - k(\boldsymbol{z}, \boldsymbol{z}')|$ is the upper bound of kernel function difference between original and latent GP.*

**Lemma 2.** *(Lemma 4-7 in (Nayebi et al., 2019)) The error upper bound of common-used kernels can be derived as follows:*

*1. For the polynomial kernel $k(\boldsymbol{x}, \boldsymbol{x}') = (\boldsymbol{x}^T\boldsymbol{x} + c)^p, l(\varepsilon) = \varepsilon pk(\boldsymbol{x}, \boldsymbol{x}')$,*

*2. For the (squared) exponential kernel $k(\boldsymbol{x}, \boldsymbol{x}') = \exp{-\|\boldsymbol{x} - \boldsymbol{x}'\|^p/c^p}, l(\varepsilon) = \varepsilon$,*

*3. For the Matérn kernel $k(\boldsymbol{x}, \boldsymbol{x}') = \frac{2^{1-\nu}}{\Gamma(\nu)}(\frac{\sqrt{2\mu}}{c}\|\boldsymbol{x} - \boldsymbol{x}'\|)^\nu B_\nu(\frac{\sqrt{2\mu}}{c}\|\boldsymbol{x} - \boldsymbol{x}'\|)$, where $\nu = p + \frac{1}{2}$ is a parameter and $B_\nu$ is a second kind Bessel function. We have $l(\varepsilon) = 2\varepsilon$.*

Using above Lemmas we can extend the probability safety guarantee from latent space to original space via additional calculations when assumptions 1 and 3 hold for original safety function $g$.

*Proof.* For simplicity we denote the upper confidence bound of safety function over original space and latent space as $u(\boldsymbol{x})$ and $\tilde{u}(\boldsymbol{z})$. Using Lemma 1, we can bound the difference when using an $\varepsilon$-subspace-embedding:

$$|u(\boldsymbol{x}) - \tilde{u}(\boldsymbol{z})| = |\mu(\boldsymbol{x}) + \beta\sigma(\boldsymbol{x}) - \tilde{\mu}(\boldsymbol{z}) - \beta\tilde{\sigma}(\boldsymbol{z})| \tag{6}$$

$$\leq |\mu(\boldsymbol{x}) - \tilde{\mu}(\boldsymbol{z})| + \beta|\sigma(\boldsymbol{x}) - \tilde{\sigma}(\boldsymbol{z})| \tag{7}$$

$$= |\mu(\boldsymbol{x}) - \tilde{\mu}(\boldsymbol{z})| + \beta\sqrt{|\sigma(\boldsymbol{x}) - \tilde{\sigma}(\boldsymbol{z})|^2} \tag{8}$$

$$\leq |\mu(\boldsymbol{x}) - \tilde{\mu}(\boldsymbol{z})| + \beta\sqrt{|(\sigma(\boldsymbol{x}) - \tilde{\sigma}(\boldsymbol{z}))(\sigma(\boldsymbol{x}) + \tilde{\sigma}(\boldsymbol{z}))|} \tag{9}$$

$$= |\mu(\boldsymbol{x}) - \tilde{\mu}(\boldsymbol{z})| + \beta\sqrt{|\sigma^2(\boldsymbol{x}) - \tilde{\sigma}^2(\boldsymbol{z})|} \tag{10}$$

$$\leq 5l(\varepsilon)\|\boldsymbol{x}\|\|X^-\boldsymbol{g}\| + 2\beta\sqrt{3l(\varepsilon)}\|\boldsymbol{x}\| \tag{11}$$

$$= (5l(\varepsilon)\|X^-\boldsymbol{g}\| + 2\beta\sqrt{3l(\varepsilon)})\|\boldsymbol{x}\|, \tag{12}$$

We denote $E(\beta, \boldsymbol{x}) = (5l(\varepsilon)\|X^-\boldsymbol{g}\| + 2\beta\sqrt{3l(\varepsilon)})\|\boldsymbol{x}\|$. Therefore, to guarantee that $\forall\boldsymbol{x}, \Pr(g(\boldsymbol{x}) > u(\boldsymbol{x})) \geq \alpha$, we need to make sure

$$\Pr(w_g(\boldsymbol{z}) \geq \tilde{u}(\boldsymbol{z}) + E(\beta, \boldsymbol{x}) \geq \alpha, \forall\boldsymbol{z} = \Pi\boldsymbol{x}, \tag{13}$$

As before, for every single $\boldsymbol{z}$, applying $r = (w_g(\boldsymbol{z}) - \tilde{\mu}(\boldsymbol{z}))/\tilde{\sigma}(\boldsymbol{z})$ and $c = \beta + E(\beta, \boldsymbol{x})/\sigma_{t-1}(\boldsymbol{z})$ for the standard normal distribution, then

$$\Pr(w_g(\boldsymbol{z}) \geq \tilde{u}(\boldsymbol{z}) + E(\beta, \boldsymbol{x}) = 1 - \Phi(\beta + E(\beta, \boldsymbol{x})/\tilde{\sigma}(\boldsymbol{z})). \tag{14}$$

In every timestep $t$, we need to set $\beta_t$ to ensure all $\boldsymbol{x} \in \mathcal{C}^X$ satisfying the probabilistic safety guarantee. Therefore , we use $\widehat{\|\boldsymbol{x}\|}$ in place of $\|\boldsymbol{x}\|$ to derive $\widehat{E}(\beta) = (5l(\varepsilon)\|X^-\boldsymbol{g}\| + 2\beta\sqrt{3l(\varepsilon)})\widehat{\|\boldsymbol{x}\|}$, which is the upper bound of $E(\beta, x)$ when fixing $\beta$. For the statement to hold, it suffice to choose $\Phi(\beta_t + \widehat{E}(\beta_t)/\widehat{\sigma}_{t-1}) \leq 1 - \alpha$.

$\square$

## B    Experimental Details

### B.1    Autoencoder training

We employ IRVAE by directly using the paper's original repository[1]. We use MLP as the VAE module for all tasks, and list the model detail in Table 1. We train all models for 300 epochs using Adam(Kingma & Ba, 2014) optimizer with a learning rate of 0.0001.

| Task | Layer number | Hiddien number | Latent dimension |
|---|---|---|---|
| Musculoskeletal model control | 2 | 512 | 5 |
| SCS simulation | 4 | 256 | 16 |
| Digital generation | 4 | 256 | 16 |

Table 1: Autoencoder model detail.

### B.2    Algorithm Implementation

For implementation of OLLSO, We use Botorch as the GP inference part(Balandat et al., 2020). We also use Botorch to replicate SCBO, CONFIG, cEI and cEI-Prob. We use the package pycma[2] to run CMA-ES on benchmarks.

All GP-based methods uses matérn kernel and fits kernel parameters after each iteration. During the experiment, we set the same trust region changing threshold for OLLSO and SCBO (the default setting of SCBO). As in SCBO and OLLSO, we use Thompson sampling as the acquisition function of CONFIG. Confidence scalar $\beta$ is set as 2 for OLLSO and CONFIG across all experiments. We set the latent optimization bound as the upper bound and lower bound of traininng points in the latent space. Other hyperparameter of the baselines are set to default values as in the original implementation.

During the experiment, we set the sample size as 10 for musculoskeletal model control task and spinal cord neuromodulation task, and sample one point each iteration in constrained digital generation task.

### B.3    Musculoskeletal Model Control

We built a full-body musculoskeletal model which actuate locomotion by controlling muscle activation (model paper under review). Here we only control the right hand part (below elbow), and fix other joints, leading to 55 muscles and 28 joints. The overall task is controlling hand muscles to hold a bottle vertically hand steadily. In the beginning of the episode, the bottle starts horizontally on the hand, and we need to first rotate the bottle and keep it in the right orientation. In each timestep, the reward from the environment is computed as follows:

$$r = r_{\text{pose}} + r_{\text{bonus}} + 10 * r_{\text{penalty}} + r_{\text{grasp}} + +r_{\text{activation}} + 100 * r_{\text{drop}} \tag{15}$$

where $r_{\text{pose}}$ is the difference between the bottle and vertical orientation, computed by Euler angle. $r_{\text{bonus}}$ is the reward when the difference below predefined threshold. $r_{\text{penalty}}$ is positive when the bottle position out of from the predefined range. $r_{\text{grasp}}$ is the distance between the centroid of the

---

[1]https://github.com/Gabe-YHLee/IRVAE-public
[2]https://github.com/CMA-ES/pycma

bottle and palm joints. $r_{\text{activation}}$ is the penalty for large muscle activations. $r_{\text{drop}}$ is the penalty when the bottle drop from hand. The overall simulation is based on MujocoTodorov et al. (2012).

When the height of the bottle is below 0.4m, we consider it has dropped from the hand and the episode is end. We record the speed of episode ending as the landing speed of the bottle. We use a safety threshold of 3.2, which is the average landing speed when randomly sampling the environment. We train a Soft Actor-Critic agent (Haarnoja et al., 2018) for 500k timesteps, and rollout for 1000 episode to build a muscle activation dataset with $71,015$ datapoints.

The performance video of OLLSOand other baselines is shown in the supplementary folder. OLLSOis capable of quickly rotating the bottle and hold vertically and steadily, while other algorithms either hold the bottle non-vertically, or learn drop the bottle safely to avoid penalty of wrong orientation.

### B.4 MODELING OF THE HUMAN SPINAL CORD

We developed an average model of the human spinal cord based on anatomical statistics(model paper under review, (Thomson, 1894; Mccotter, 1916; Zhou et al., 2010; Rowald et al., 2022; Kameyama et al., 1996)). The model contains gray matter, white matter, nerve roots, cerebrospinal fluid (CSF), and dura mater of T12-S2 segments of the spinal cord which are related with the motor control of lower limbs. The specific conductivity values of the modeled tissues were set refer to (Ladenbauer et al., 2010). Electric fields induced by different stimulation parameters were derived using finite element method (FEM). To calculate the stimulation effects for different muscles, we redistricted the cord model according to reported results of the segmental innervation for lower limb muscles((Sharrard, 1964; Schirmer et al., 2011)). Six groups of muscles of bilateral lower limbs were studied: iliopsoas (IL), vastus lateralis and rectus femoris (VL&RF), tibialis anterior (TA), biceps femoris muscle and gluteus maximus (BF&GM), semitendinosus (ST), and gastrocnemius (GA).

To evaluate the selectivity of stimulation for certain muscle, we used a selectivity index (SI) to characterize the distribution of the electric field. The selectivity index for the $i$th muscle was defined as follows:

$$\text{SI}_{\text{i}} = \mu_i - \frac{1}{m_{\text{neighbor}} - 1} \sum_{j \neq i}^{m_{\text{neighbor}}} \mu_j \tag{16}$$

where $m_{\text{neighbor}}$ represents the number of muscles whose motor neuron pools are adjacent to the $i$th muscle's. The selectivity index ranges from -1 to 1, where -1 represents the maximum of activation of all undesired muscles with a complete absence of activation of the targeted muscle, 0 indicates that all muscles are activated at the same level, and 1 means the targeted muscle is activated at the greatest extent while no undesired muscles are activated. And $\mu_i$ is the normalized activation of the $i$th muscle and is defined as follows in the simulation:

$$\mu_i = \frac{\iiint_{\Omega_i} f(x,y,z) \mathrm{d}x\mathrm{d}y\mathrm{d}z}{\iiint_{\Omega_i} 1 \mathrm{d}x\mathrm{d}y\mathrm{d}z} \tag{17}$$

$$f(x,y,z) = \begin{cases} 1, & \text{if } AF(x,y,z) > AF_{\text{threshold}} \\ 0, & \text{if } AF(x,y,z) \leq AF_{\text{threshold}} \end{cases} \tag{18}$$

$\Omega_i$ is the segmental volume of the $i$th muscle in the cord. $AF$ is the activating function, defined as the second spatial derivative of extracellular voltage along an axon((Rattay & Frank, 1986; Butson & McIntyre, 2006)).

We use the spinal model to traverse all stimulation parameters with 1 cathode with anodes no more than 3, and 2 cathodes with anodes no more than 2, leading to a SCS dataset with 218,000 stimulation parameters and predicted muscle activation. We compute the objective function using 16, and compute the safety function as $g(x) = 1 - max_i(\mu_i)$. The selectivity index distribution is shown in Figure 5. We set the safe threshold as 0.05, with nearly half of the traversed parameters are safe.

We convert electrode parameters from 17d vector to a 2d electric field image using simplyfied computation. In concrete, we map contact combinations to the spatial position in the electrode, linearly

compute the diffusion of electrical field from each cathode and anode, and multiply the map by current intensity. One example of generated electric field image is shown in Figure 6.

We use the constructed SCS dataset as the function oracle. For a given 2d map, we set its function value as the function value of the nearest unevaluated point in the dataset.

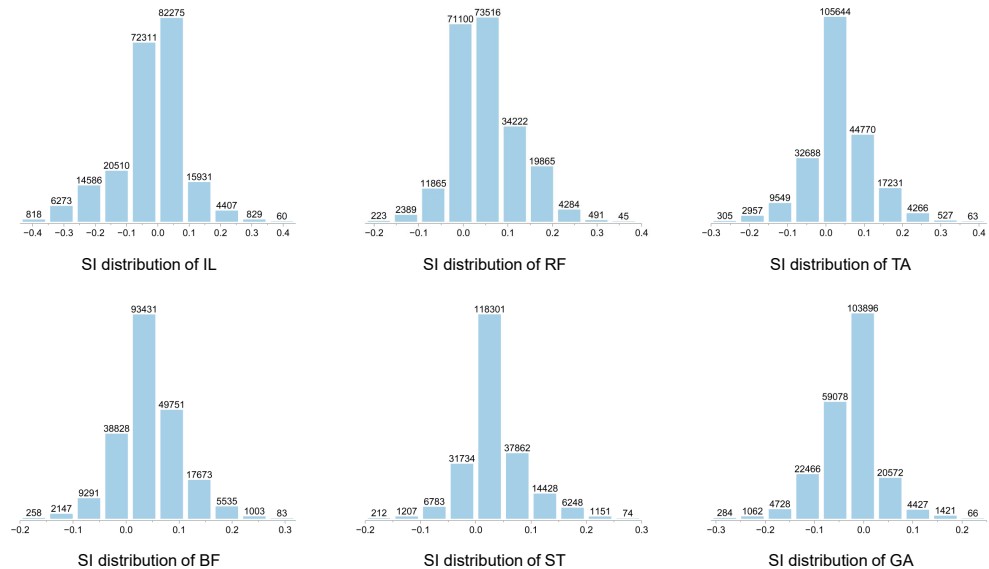

Figure 5: Distribution of SI for six muscle groups of different configurations used in SCS simulation experiment

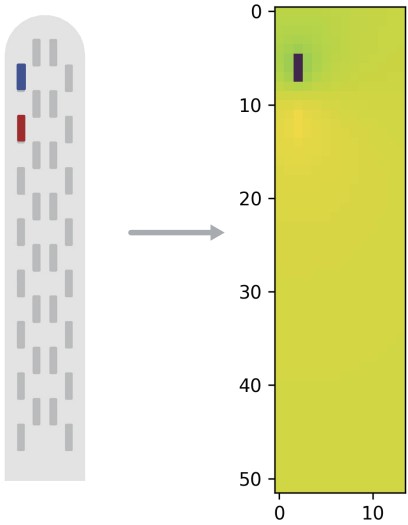

Figure 6: 2d electrical map computation.

## B.5    CLINICAL EXPERIMENT OF SPINAL CORD STIMULATION

We employ OLLSO in the treatment of spinal cord stimulation to find more selective stimulation parameters for different muscles. All the trials were conducted under the supervision of therapists. The patient was seated in the wheelchair in a comfortable way and was told to relax. At the first period, typical parameters which were usually used in the therapy (e.g. bipolar stimulation) were delivered to the patient while the evoked muscle activities were recorded using EMG. These data

(132 trials) were used to initialize `OLLSO`. Except for the first 132 trials as the initial data, 441 out of 504 trials are recommended by `OLLSO`. The other trials were conducted by the therapist.

We focus on 8 groups of muscles: iliopsoas (IL), rectus femoris (RF), tibialis anterior (TA), and biceps femoris (BF) for both sides. The clinical selectivity index was defined as following:

$$\text{SI}_i = \frac{\mu_i}{1 + \sum_{j \neq i}^{m} \mu_j} \tag{19}$$

where $\mu_i$ represents the normalized peak-to-peak value of the evoked EMG for the $i$th muscle and $m$ is the total number of the target muscles.

For each trial, our algorithm recommended the parameter based on the history data and configured it onto the stimulator, which would deliver electrical stimulation to the patient for 800 ms at a frequency of 10 Hz. Peak-to-peak values were averaged and normalized to obtain selectivity indices of different muscles after stimulation. The calculated feedback and queried safety score were used to update the optimizer and it would recommend a new parameter. During the optimization process, the therapist also provided a small part of parameters for different targets (63 out of 504 trials, non-shadow area in 3 (b)). The tasks were refined sequentially and all the history data were reused when optimizing a new task.

### B.6 DISTANCE PRESERVING OVER LATENT SPACE

To demonstrate the distance preserving capability of IRVAE. We first randomly sample 10,000 points from the muscle activation dataset and compute the pair-wise distance in original and latent space, shown in Figure 4 (a).

Then we use a GP with Matérn kernel (lengthscale=1 for safety function and 5 for utility function) to prediction means and variance of the safety function given policy parameters. The training data is the initial original and latent policy parameters. The testing data is 36000 randomly chosen original policy parameters, the GP mean and variance esimation of orignal and latent GP is shown in in Figure 4 (b)-(c).

We further compute point-wise distance of 10,000 random sampled points. The results shows high linear correlation of point-wise distance between original space and latent space ( is 0.851 for musculoskeletal model control task and 0.973 for neural stimulation task), indicating good distance preserving ability of IRVAE.

## C    ADDITIONAL EXPERIMENT

### C.1    HAND-WRITING DIGITAL GENERATION

In hand-writing digital generation task, the goal is to generate images of target digit as thick as possible, while keeping the image valid for the required number. Using this task we can test the algorithm performances when the latent dimension is low. We trained a fully-connected IRVAE with a latent dimension of 6 and use a two-layer CNN model as the predictor. We set the objective function as the sum of image pixel intensities and the safety function as the prediction probability of target number. Since the CNN prediction is very sharp, we wrap the CNN output via a softmax layer with temperature as 200. We set the sample budget as 200 including with 20 images of target digit as the initial data.

We show the optimization result of generating number 0-9 figure 7 and summarize the averaging perfromance in Table 2. We observed `OLLSO` outperforms all baselines in terms of optimization performance and safety violation. Note that while `OLLSO` efficiently finds highest objective, its safety violations is 63% less than the second best method original SCBO.

### C.2    ABLATION ON CONFIDENCE BOUND SCALAR

Here we run `OLLSO` with $\beta = 0, 2, 4, 6, 8$ on musculoskeletal model control task, and shown the results in Figure 8. We observe the algorithm performance is similar under a wide range choice of $\beta$ on this high-dimensional task.

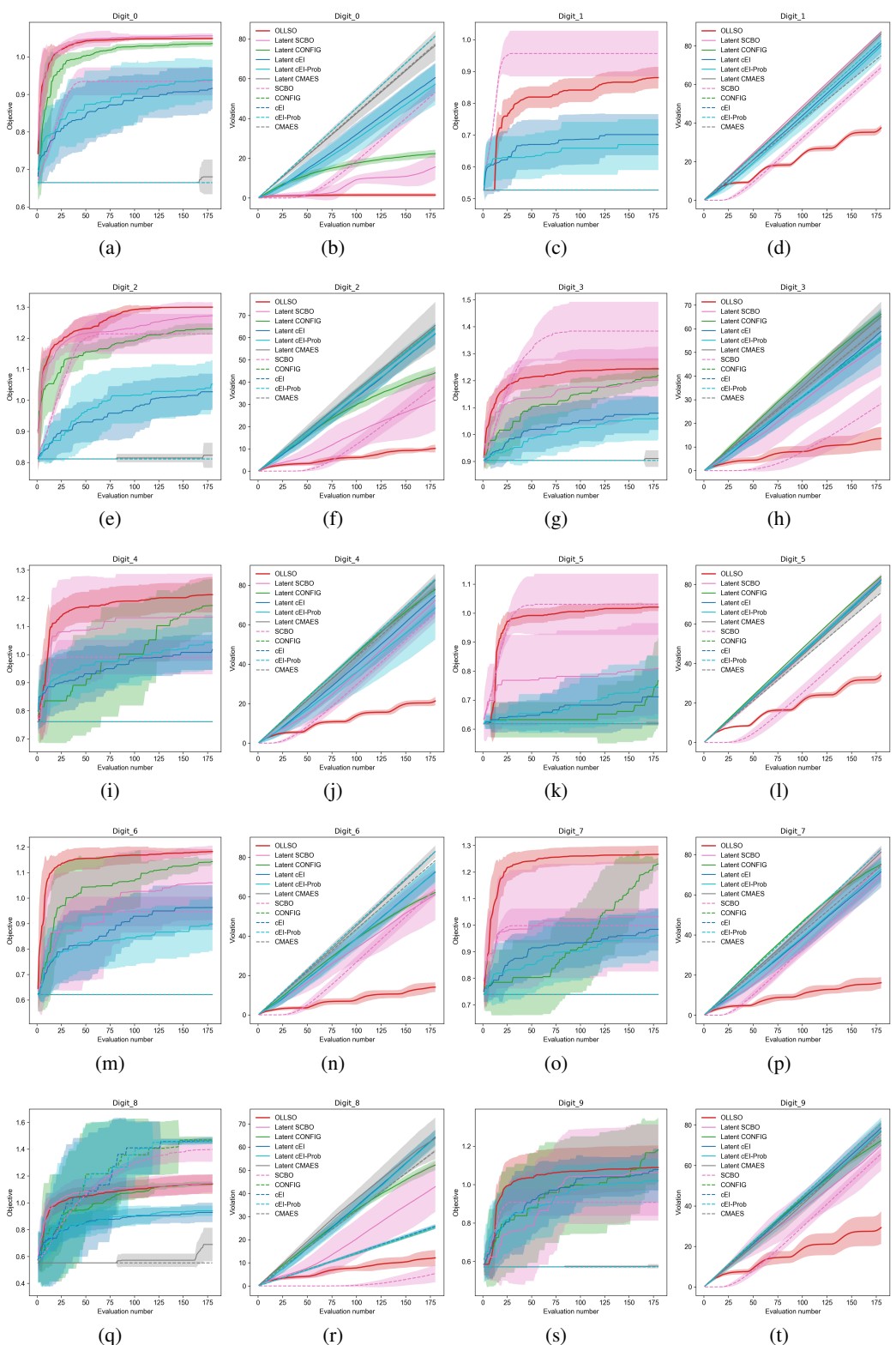

Figure 7: Experiment of hand-writing digital generation of number 0-9.

Table 2: Experiment results of constrained hand-writing digital generation. We evaluate algorithm performance of generating digital from 0 to 9 in terms of best found feasible objective value (higher is better) and cumulative safety violation (lower is better). Objective values are normalized by best feasible point in the MNIST dataset. The results are shown as mean performance $\pm$ one standard deviation across ten tasks.

| Metric | Method | OLLSO | SCBO | CONFIG | cEI | cEI-Prob | CMAES |
|---|---|---|---|---|---|---|---|
| Objective | Latent | **1.14 ± 0.12** | 1.03±0.2 | 1.07±0.22 | 0.94±0.13 | 0.93±0.13 | 0.69±0.11 |
|  | Original |  | 1.08±0.18 | 0.77±0.26 | 0.77±0.26 | 0.77±0.26 | 0.68±0.12 |
| Violation | Latent | **18.99 ± 10.82** | 59.25±21.97 | 64.39±18.99 | 70.6±8.32 | 69.29±8.69 | 76.15±7.88 |
|  | Original |  | 52.01±20.28 | 72.09±17.85 | 72.04±17.82 | 72.0±17.87 | 72.01±7.06 |

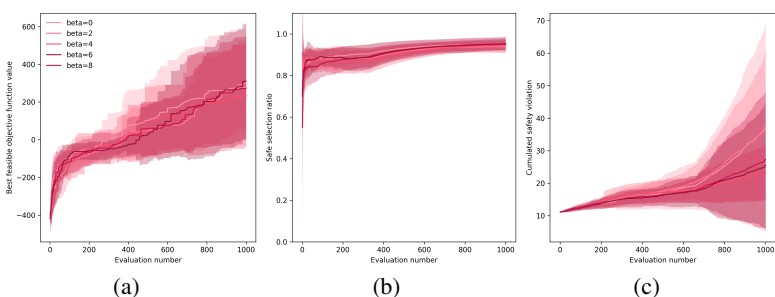

|  | (a) | (b) | (c) |
|---|---|---|---|

Figure 8: Ablation study on confidence bound scalar $\beta$.

### C.3 COMPARISON WITH RANDOM-EMBEDDING BO

We additionally run HesBO and BAxUS on the musculoskeletal model control task and the neural stimulation task with no safety constraints in simulation. Due to the algorithmic mechanism of HesBO and BAxUS, we cannot directly use the same initial point as OLLSO. Therefore we randomly sample initial points from their corresponding latent space. In HesBO, we set the same latent dimension number as in OLLSO. Table 3 shows the best objective function values found by algorithms (shown as mean $\pm$ 1 std).

We observe OLLSO still outperforms HesBO and BAxUS across all tasks, even when optimizing under safety constraint. We think using IRVAE enables utilizing the pre-collected unlabelled data to learn a better representation than random projection.

## D ADDITIONAL RELATED WORK

Here we additionally discuss more works about high-dimensional Bayesian optimization besides dimension-reduction based BO and local BO.

Due to the inversion of kernel function matrix, the complexity of GP inference scales exponentially with the sample number, limiting the search budget of high-dimensional problems. Sparse GP or variational GP is used to achieve scalable sampling over the high-dimensional space (Seeger et al., 2003; Snelson & Ghahramani, 2005; Hensman et al., 2013).

Table 3: Best objective function values found by algorithms.

| Algorithm | Muscle | SCS-IL | SCS-RF | SCS-TA | SCS-BF | SCS-ST | SCS-GA |
|---|---|---|---|---|---|---|---|
| OLLSO | **284.08 ± 305.95** | **0.39 ± 0.03** | **0.37 ± 0.00** | **0.38 ± 0.02** | **0.28 ± 0.03** | **0.26 ± 0.02** | **0.22 ± 0.01** |
| HesBO | −747.8 ± 124.53 | 0.34 ± 0.03 | 0.30 ± 0.04 | 0.36 ± 0.05 | 0.25 ± 0.01 | 0.23 ± 0.0 | 0.19 ± 0.02 |
| BAxUS | −619.36 ± 302.83 | 0.36 ± 0.04 | 0.35 ± 0.02 | 0.34 ± 0.05 | 0.27 ± 0.02 | 0.24 ± 0.03 | 0.19 ± 0.05 |

Another line of work assume the addictive structure of the objective functions, and decomposes the function to solve the low-dimensional sub-problem decentrally(Kandasamy et al., 2015; Gardner et al., 2017; Wang et al., 2018; Mutny & Krause, 2018).

To overcome over-exploration issue over the high-dimensional space, several works also propose shape kernel prior to sample points near the search center (Oh et al., 2018; Eriksson & Jankowiak, 2021).

## E  FUTURE WORKS

While our current work train VAEs in a unsupervised manner, we wonder whether the optimization could be more safe and efficient when shaping the latent space and GP prior with collected labelled data. We are also interested in analyzing the safety guarantee under noisy observation setting and try to employ OLLSO to safely optimize other real-world problems in the future.

