# OpenReview forum: "High-Dimensional Safe Exploration via Optimistic Local Latent Safe Optimization"
_ICLR.cc/2024/Conference — Submitted to ICLR 2024_

### Official Review · Reviewer_Hhpv · 2023-10-25

**Soundness:** 3 good
**Presentation:** 2 fair
**Contribution:** 2 fair
**Rating:** 8
**Confidence:** 3

**Summary:**

This paper introduces the Optimistic Local Latent Safe Optimization (OLLSO) method, designed to address the challenging problem of optimization in high-dimensional input spaces while maintaining safety during sampling. OLLSO leverages a distance-preserved autoencoder to transform the original high-dimensional input space into a continuous latent space, allowing for efficient optimization of utility functions. The paper provides a theoretical probabilistic safety guarantee from the latent space to the original space. In simulation experiments, OLLSO outperforms existing high-dimensional constrained optimization algorithms. Furthermore, the paper demonstrates OLLSO's real-world application in clinical experiments, where it efficiently and safely optimizes neuromodulation therapy for a paraplegic patient, showcasing its practical utility in safety-critical contexts. Overall, OLLSO offers a promising approach for addressing high-dimensional optimization problems while ensuring safety, with potential applications in various fields.

**Strengths:**

The paper's strength lies in its introduction of the Optimistic Local Latent Safe Optimization (OLLSO) method, tailored for addressing safety concerns in high-dimensional sequential optimization problems. OLLSO's innovation comes from its utilization of a regularized autoencoder to transform complex, high-dimensional input spaces into continuous latent spaces while preserving distances, enabling efficient handling of hybrid inputs. The algorithm then employs an optimistic local safe strategy within the latent space, distinguishing safe regions through the upper confidence bounds of the safety function. Furthermore, the paper provides a rigorous theoretical probabilistic safety guarantee that spans from the latent space to the original input space. OLLSO's practicality is demonstrated through its application to safety-critical scenarios, where it outperforms existing high-dimensional constrained Bayesian optimization algorithms in terms of both optimization efficiency and safety during sampling. Its successful deployment in real clinical experiments, optimizing lower limb muscle control for a paraplegic patient, underscores OLLSO's potential for impactful real-world applications in safety-critical domains.

**Weaknesses:**

- The assumptions limit the widespread application of the proposed algorithm.
- The presentation has the potential to be improved, especially Section 4.1.
- The contributions could be better summarized.

**Questions:**

- The presentation in Section 4 should be better polished. In my mind, the definitions of the acquisition function A and the local region are missed at least. It would also be helpful if the authors could make a plot to show what they (side length, local region...) are.
- Considering the mapping from the original state space to the latent space is a key contribution, the author should explain more about why IRVAE is better in 4.1, and indicate where the related experimental results are.
-  Considering the assumptions, it would be quite helpful if the authors could have a more general description of how we can apply the methods in the real world. Besides, why does the real clinical therapy optimization in Section 6.1 satisfy the assumptions?
- After Equation (1), I do not think $\alpha$ smaller than 0.5 will make the solution safe. Maybe I misunderstand some things. In Algorithm 1, there is an extra half bracket in the line 9.
- Even though the work is quite different from the safe exploration in RL [1], it would be great if the authors could have some discussion about their differences in terms of problem setting, safety mechanism, etc. The researchers in Safe RL would quite appreciate these discussions.

[1] Yang Q, Simão T D, Jansen N, et al. Reinforcement Learning by Guided Safe Exploration, ECAI 2023.

---

> ### Author Response · Authors · 2023-11-17
> **Rebuttal [1/2]**
>
> Thanks for your encouraging comments of our work, and we respond your questions below.
>
> **The assumptions limit the widespread application.**
>
> We argue that the most assumptions we used in the paper are common-used in high-dimensional BO literature. Assumptions 1 and 2 are regularity assumptions of the function, which enables to use Gaussian process to model function distributions. Assumption 3 is low effective dimension assumption common-used in embedding BO works to manifest the structural reward characteristic. Note the low-dimensional manifold are always observed in most high-dimensional structures. The only additional assumption is the distance preserving ability of the latent space embedding, which is crucial to guarantee probabilistic safety and can be satisfied using some linear embedding, or approximately satisfied using isometry regularized VAE. We have demonstrated success of our proposed method in two different complex high-dimensional problems and one real-world application. And we expect to apply OLLSO to more real-word safety-critical problems.
>
>
>
>
> **General recipe of applying OLLSO in real-world applications.**
>
> We believe OLLSO can be applied in various high-dimensional safety-critical problems. As mentioned above, the regularity assumptions are common-used in BO literature. In practice, we can fit the kernel hyperparameter using the observed data to make GP better model the functions. Many high-dimensional problems often have low-dimensional intrinsic structure, which satisfy the low effective dimension assumption. The distance preserving assumption depends on the embedding way, where using a IRVAE is able to approximately satisfy the condition.
>
> In the real clinical therapy optimization, most dimensions (16 contact configurations) of the original input space are discrete, which is a sparse representation, and has the potential to reduce to a more compact representation. Our experimental result of IRVAE also shows that it can well preserve the distance and reconstruct the original input. Note that this exam of distance preserving is conducted before real evaluation, and we can start the real experiment after ensure the embedding satisfy the distance preserving requirement.
>
> Overall, to apply OLLSO, the first step is to train an IRVAE using pre-collected or synthesized unlabelled data without actual evaluation. After verifying the embedding distance preserving capability, we can safely optimize the objective function following Algorithm 1. It is easy to be applied to other real-world problems.
>
> **Safety probabilistic threshold.**
>
> Thank you for pointing out the typos. We will fix them in the next version.
>
> In many real-world applications, we can exclude a large number of unsafe positions using prior knowledge. The remaining search region may have a certain probability to be safe even under random search.
> Therefore we set the probabilistic threshold smaller than 0.5, which is an optimistic estimation of the safety function in the problem formulation, and enables to use UCB to optimistically identify the safe region. We believe this formulation offers a more practical way to efficiently and safely optimize high-dimensional real-world problems.
> Our experimental result also shows that OLLSO usually achieves safety probability larger than 80\% in the whole optimization procedure, and significant less constraint violation compared to other competitive baselines, which empirically demonstrates the safety guarantee when applying OLLSO in the real-world applications.
>
> **Discussion with safe exploration in RL**
>
> Thank you for listing the related work about safe exploration in RL.
> We will include the discussion of safe exploration in RL [2] in the next version.

---

> ### Author Response · Authors · 2023-11-17
> **Rebuttal [2/2]**
>
> **The presentation has the potential to be improved and the contributions could be better summarized**
>
> Thank you for your suggestion, and we will improve the presentation in the next version.
>
> Here we summarize our contribution to as follows:
>
> 1. In this paper, we propose a practical method to achieve safe and efficient optimization of high-dimensional functions. To our best knowledge, no existing work focuses on guaranteeing safety during the high-dimensional optimization.
>
> 2. OLLSO overcomes the scalability and efficiency difficulties of the previous GP-based safe optimization method by introducing a local search mechanism with optimistic safety region identification, and achieves safe and efficient optimization of functions with hundreds of dimensions.
>
>  3. OLLSO is the first algorithm to use distance-preserved VAE to address safety issue in latent space optimization. We derive the theoretical probabilistic safety guarantee of OLLSO, which enable it to handle extremely high-dimension or discrete input space.
>
>  4. Our experimental results shows that OLLSO significantly outperforms existing safe BO methods, constrained BO algorithms, and direct combinations with dimension reduction models.
>
> We want to emphasize that OLLSO is a PRACTICAL method that can safely optimize high-dimensional real-world problems. In addition to state-of-the-art simulation performance, our real clinical experiments show that OLLSO successfully optimizes neural stimulation for the control of human muscles. While the majority of our cited papers conduct their experiments only in simulation or synthetic problems, OLLSO is both practical and theoretically verifiable to safely and efficiently optimize real-world safety-critical problems.
>
>
>
> **Methodology clarification.**
>
> We describe the choice of the acquisition function $A$ and the updates of the local region in Section 4.1.
>
> For the acquistion function $A$, we use Thompson sampling[1] as the acquisition function due to its compatibility with the discrete nature of our safety estimation and search space, and its innate ability for batch optimization by sampling the GP posterior.
>
>  The local region $L_t$ is a trust region defined by a center (using current best feasible point) and a side length $l$, where we dynamically update the search space based on the sample history. Thank you for pointing out the confusion part and we would improve the presentation and add some illustration figures or psudo-codes to help readers better understanding our paper.
>
> **Further analysis about IRVAE.**
>
> We show the experiment of analyzing the IRVAE in Section 6.3. And we will improve our presentation in Section 4.1 to lead readers to the experiment part.
>
> In Figure 4, we show that when using IRVAE, the GP estimation difference between original and latent space is significantly smaller than using standard VAE. We further compute point-wise distance of 10,000 random sampled points. The results shows high linear correlation of point-wise distance between original space and latent space ($R^2$ is 0.851 for musculoskeletal model control task and 0.973 for neural stimulation task), indicating good distance preserving ability of IRVAE.
>
> [1] Kandasamy, Kirthevasan, et al. "Parallelised Bayesian optimisation via Thompson sampling." International Conference on Artificial Intelligence and Statistics. PMLR, 2018.
>
> [2] Yang Q, Simão T D, Jansen N, et al. Reinforcement Learning by Guided Safe Exploration, ECAI 2023.

---

> > ### Comment · Reviewer_Hhpv · 2023-11-18
> >
> > Thanks to the authors for their response.
> >
> > In general, my concerns were addressed. I am happy to raise my score.

---

> > > ### Author Response · Authors · 2023-11-20
> > >
> > > Thanks for your appreciation! We will add more instructions for easier use when we release the code repository.

---

### Official Review · Reviewer_N9rb · 2023-10-29

**Soundness:** 1 poor
**Presentation:** 2 fair
**Contribution:** 1 poor
**Rating:** 3
**Confidence:** 5

**Summary:**

The paper proposes Optimistic Local Latent Safe Optimization (OLLSO), a method for safe exploration in high-dimensional search spaces. The difficulty of high-dimensional Gaussian Process (GP) modeling is circumvented through the use of a distance-preserving variational autoencoder (VAE). Modeling is performed in the low-dimensional latent space produced by the VAE. Theoretical analysis is provided on the probabilistic safety of the algorithm. Lastly, OLLSO displays improved empirical performance over competing algorithms, and justifies the choice of VAE through ablations.

**Strengths:**

__A:__
__Figure 1:__ A very nice illustration of the proposed algorithm, which effectively summarizes it in a pedagogical manner.
__Figure 4:__ A welcomed ablation on the choice of model.
__Impressive results:__ OLLSO yields good results on all tasks considered.

**Weaknesses:**

- __Probabilistic safety:__ Proposed as a novel concept, I find it confusing and believe it needs to be discussed further.

Specifically, we seek to find the optimal value $x^* = \max f(x)$ under the condition that $x^*$ is safe, __probably__. Do we not observe whether $x^*$ is actually safe? Is the safety at $x^*$, or any other $x$ we have observed, stochastic? Since I have not completely understood this central aspect of the paper, I reserve judgement until it is further clarified, but the objective in Eq. 1 does not currently appear to be properly defined.

- __Assumptions of unclear validity & subsequent theorem:__
  - __Assumption 1:__ While this is a common assumption, _it is conventionally made on the original input space_. Assuming that the VAE latent space adheres to this assumption would entail that the projections from $\mathcal{Z}$ are sufficiently well-behaved. In vanilla BO, the modeling choices adhere to the assumption through the use of an appropriate kernel (Matern, RBF). However, it is not clear whether the decoding step is well-behaved (and due to its distortion of the space, I would be surprised if if was), so that _the assumption is adhered to_ in the proposed setup. I suggest the authors re-visit this assumption, since its correctness hinges on the continuity of the decoder. What makes them believe that the norm-bound $B$ is preserved through the decoding step? Since the theoretical analysis hinges on Assumption 1, which is decidedly _not_ common in latent space BO, the gap between the conventional assumption and the one made in latent space should be theoretically justified.

  - __Assumption 3:__ The mapping $U$ is not currently properly defined, since it does not have an output. I believe you mean that $U$ is a _matrix_ or a map $R^D --> R^{d_e}$ (as in HeSBO). Further, wouldn't $g(\mathbf(x))$ imply that $g$ accepts D-dimensional input and $g(\mathbf(Ux))$ imply that $g$ accepts $d_e$-dimensional input?
  - __Subsequent subspace assumption (After A3, henceforth AA3):__ _"Here we assume that the latent space mapping is an ε-subspace mapping "_. Please re-state this as a proper assumption for clarity. Moreover, is there a fundamental reason to believe this holds true, i.e. can properties of the IRVAE shed light on this? In HeSBO, the construction of the algorithm ensure this holds true. I don't see why the VAE would satisfy this unless it employed only linear layers (which I assume it does not).
  - Lastly, the intermediary conclusion, _"we can properly choose $\beta_t$ that satisfies the probability safety guarantee, under the noise-free setting"_ does not mesh well with __Assumption 2__ - observations are perturbed by noise.

Due to the three (A1, A3, AA3) seemingly unverifiable and seemingly incorrect (A1, AA3) assumptions that are made, I do not attribute any value to the theory that is being presented. I encourage the authors to shed light on why these assumptions are valid.

- __Novelty of method:__ OLLSO combines exisiting VAE architectures with trust regions, Thompson sampling and constaints. VAEs for BO have been extensively explored, and the other components are exactly SCBO (Eriksson and Poloczek 2021). While the combination is interesting, no novel building blocks for BO are proposed.

- __Brief methodology section:__ Section 4. is brief and too high-level. Thus, it leaves a few questions unanswered. These are all outlined in the _Questions_ below. I believe addressing these would add clarity to the paper. Moreover, Section 4.2 is related work and should, in my opinion, be moved to Section 2.

- __VAE-related background:__ BO and GPs are well-covered in the background, but the VAE-related design choices are not. As it is a vital part of the algorithm, background on the IRVAE is integral to the paper.

- __Parsing of results:__ Plots are difficult to process due to the small fontsize, and text which overlaps with the rest of the plot (Fig. 4). Consider making it substantially larger, possibly at the expense of moving some of it to the Appendix.

__Minor:__
- _"estimates both the posterior and the confidence interval"_. The confidence interval comes for free after estimating the posterior, so saying that the posterior is estimated suffices.
- _"Follows the standard local Bayesian optimization algorithms ..."_ --> Following
- $u_t(z)$: If the $\max C$ is over $z$ (which it should be) then  $u_t$ is not a function of $z$.
- Fig. 1: Optimisticc --> Optimistic

**Questions:**

__Should be clarified:__
- _"Our approach resets l to its initial length, ensuring a different, safer trajectory sampling than the initial instance."_ Why does this action _ensure_ a safer trajectory?
- Algorithm 1, line 2: What is the _trajectory_ $\zeta_0$?
- For the trust region, how is $\ell$ set?

__General questions:__
- Why can unlabeled data be synthetized (in large quantities)?

---

> ### Author Response · Authors · 2023-11-17
> **Rebuttal [1/2]**
>
> Thanks for your detailed feedback on our work, and it greatly helps us improve our paper. We answer your questions below.
>
> **Optimization over probabilistic safety constraints**
>
> Thank you for pointing out the confusion in the problem formulation. We actually want to find an optimal $x^*$ with the highest feasible objective function value, whose safe value is above the pre-defined threshold after observation. During the optimization procedure, we want every decision we make to be probabilistic safe. We will further modify our problem formulation in the next version to make it more clear, but it does not influence the final theoretical result, where we prove that every decision we made during optimization satisfies the probabilistic safe requirement of the problem.
>
> **Regularity assumption over latent space.**
>
> Our regularity assumption over the latent space enables to use GP to model function over the latent space. We agree that the norm-bound can not be preserved over the original space when the decoder is not well-behaved. Therefore, we want to use an isometric regularized autoencoder to achieve approximated distance preserving over the latent space. Our theoretical result(Theorem 1) is built upon both the regularity assumption(Assumption 1) and the distance preserving ability(Definition 1) of the latent embedding.
>
> **Assumption about the effective dimension.**
>
> We apologize for the not explicitly defining the low effective dimension of the original function. The correct formulation should be
>
> $g(\boldsymbol{x})=g(\boldsymbol{z}U^T)$
>
> where $\boldsymbol{z}=U\boldsymbol{x}$, and $U^T$ is the inverse mapping from latent space to the original space.
>
> **$\epsilon$-subspace embedding assumption**
>
> We agree that the $\epsilon$-subspace embedding assumption may not hold for standard VAE with non-linear layers. But we argue that IRVAE may approximately satisfied this assumption. In figure 4, we show that the GP estimation difference between original space and latent space of IRVAE is significantly smaller than latent space of standard VAE.
> We also compute point-wise distance of 10,000 random sampled points. The results shows high linear correlation of point-wise distance between original space and latent space ($R^2$ is 0.851 for musculoskeletal model control task and 0.973 for neural stimulation task), indicating good distance preserving ability of IRVAE.
>
> **Assumptions about noise observation.**
>
> We propose OLLSO to optimize general safety-critical problems with noisy observation. In this paper, we theoretically derived the safety guarantee under noise-free setting, and we believe under bounded noise assumption, the derivation of safety guarantee under noise-perturbed observation is feasible.
>
> We also empirically demonstrate that OLLSO works well under noisy setting in the clinical experiment, where the observation is perturbed by noise from measuring EMG signal of lower limb muscles.
>
> **Methodology clarification.**
>
> 1. Explanation of safer trajectory after reset.
>
> When resetting the trust region, we preserve all sample points compared to SCBO, which discards all history data. In this way, we have more knowledge about the safety function compared to last reset point, and the future sample procedure will be safer.
>
> 2. The sample trajectory
>
> The sample trajectory $\zeta_{t}$ is defined as a set of tuples {$(\boldsymbol{x}_t,y_t^f,y_t^g)$}. The update of the trust region is based on the selection result of last round, which is a process depending on the historical selection trajectory.
>
> 3. The setting of trust region length $l$
>
> We directly use the setting of SCBO BoTorch implementation(https://botorch.org/tutorials/scalable_constrained_bo). In concrete, we set the initial $l$ as 0.8, $l_{min}$ is $0.5^7$, and $l_{max}$ is 1.6. We normalize all inputs in the range of $[0,1]$, therefore we can use same setting across different tasks.
>
> 4. The upper confidence bound
>
> We define $u_t(\boldsymbol{z})\coloneqq\max C_t(\boldsymbol{z})$, where $C_t(\boldsymbol{z})\coloneqq [\mu_t(\boldsymbol{z})\pm\beta\sigma_t(\boldsymbol{z})]$ is the confidence interval at the position $\boldsymbol{z}$. Therefore, the max operation is performed over the interval instead of $\boldsymbol{z}$.
>
> **VAE-related background**
>
> Thank you for pointing out the missing discussion of this topic. We will include it in the next version of the paper.
>
> **Unlabeled dataset synthesizing**
>
> Taking neural stimulation task as an example, we synthesize the unlabelled stimulation parameter by randomly setting the contact configuration and the intensity under clinical prior (e.g. limiting the contact number or intensity range). The procedure is capable of generating large quantity of data without actual evaluation. There are many other real-world applications where we can synthesize data in the same way, such as protein/molecular sequences.
>
> **Parsing of results and typos**
>
> Thank you for finding typos in our paper. We will correct them in the next version.

---

> ### Author Response · Authors · 2023-11-17
> **Rebuttal [2/2]**
>
> **The novelty of the method.**
>
> In this paper, we propose a practical method to achieve safe and efficient optimization of high-dimensional functions. To our best knowledge, no existing work focuses on guaranteeing safety during the high-dimensional optimization.
>
> We present the novelty of OLLSO in both algorithmic and practical aspects.
>
> In terms of latent space optimization, previous latent BO works neglect the geometry-preserving during unsupervised representation learning, which is crucial to extend safety guarantee from latent space to the original space. Many of them achieve latent space shaping using collected labeled data [1, 2]. We are the first to use distance-preserved VAE to address safety issue in latent space optimization, and derive the theoretical probabilistic safety guarantee of our proposed method.
>
> The local region adaptation of OLLSO is also different from existing
> trust-region based BO methods[3, 4] which neglect the safety during the optimization procedure. They would also discard all previous sample point when restarting the trust region. OLLSO updates the trust region in a more safety-sensitive way, and keeps all previous samples when restarting the trust region, ensuring a safer optimization procedure with theoretical guarantee.
>
> In terms of the safe optimization part, previous GP-based safe optimization method are limited to low-dimensional problems (typically below 10, [5, 6]). OLLSO addresses this issue by introducing local search mechanism upon optimistic safety region identification, and is able to handle problem with hundreds of dimensions.
>
> Our experimental result also shows that OLLSO is not a direct application to comprise exsiting algorithms. Figure 2 and Figure 3 show that simply combining IRVAE with existing safe BO (CONFIG (L)) or trust region-based constrained BO (SCBO (L)) perform worse than OLLSO in terms of both optimization performance and safety guarantee. Our simulation results shows that OLLSO significantly outperforms existing safe BO methods, constrained BO algorithms, and direct combinations with dimension reduction models.
>
> We want to emphasize that OLLSO is a PRACTICAL method that can safely optimize high-dimensional real-world problems. In addition to state-of-the-art simulation performance, our real clinical experiments show that OLLSO successfully optimizes neural stimulation for the control of human muscles. While the majority of our cited papers conduct their experiments only in simulation or synthetic problems, OLLSO is both practical and theoretically verifiable to safely and efficiently optimize real-world safety-critical problems.
>
> [1] Tripp, Austin, Erik Daxberger, and José Miguel Hernández-Lobato. "Sample-efficient optimization in the latent space of deep generative models via weighted retraining." Advances in Neural Information Processing Systems 33 (2020): 11259-11272.
>
> [2] Grosnit, Antoine, et al. "High-dimensional Bayesian optimisation with variational autoencoders and deep metric learning." arXiv preprint arXiv:2106.03609 (2021).
>
> [3] Eriksson, David, et al. "Scalable global optimization via local Bayesian optimization." Advances in neural information processing systems 32 (2019).
>
> [4] Eriksson, David, and Matthias Poloczek. "Scalable constrained Bayesian optimization." International Conference on Artificial Intelligence and Statistics. PMLR, 2021.
>
> [5] Sui, Yanan, et al. "Safe exploration for optimization with Gaussian processes." International conference on machine learning. PMLR, 2015.
>
> [6] Xu, Wenjie, et al. "Constrained efficient global optimization of expensive black-box functions." International Conference on Machine Learning. PMLR, 2023.

---

> ### Comment · Reviewer_N9rb · 2023-11-17
> **Intermediate response**
>
> Thanks to the authors for their response. I wIll require more time to make a judgement on the full response, but I believe one point need to be addressed urgently:
>
> Authors' response:
> __Optimization over probabilistic safety constraints__
>
> I am unfortunately not satisfied with the response. As clarity of the problem at hand is paramount to any paper, I wish to further understand it to make an informed judgement:
>
> ##### _We actually want to find an optimal with the highest feasible objective function value, whose safe value is above the pre-defined threshold after observation._
>
> So if I understand correctly, the safety value $g(x)$ of any configuration is observed in conjunction with the objective $f(x)$ ? If so, why is the safety $g$ probabilistic? (since we know if $g(x) > h$ for sure once we have observed it, hence there is no probabilistic component).
>
>
> I encourage the authors to be precise and update the paper on this point.

---

> > ### Author Response · Authors · 2023-11-18
> >
> > Thank you for your quick reply!
> >
> > In our general problem setting, the safety function is also an unknown function (with bounded norm in a Reproducing Kernel Hilbert Space) with observation noise. Sampling at the same point will induce a noisy observation. In the specific noise-free setting, since we do not know the actual safety function, the output of the given input is also a stochastic distribution estimated by the surrogate model. Therefore, the uncertainty before making the actual decision consists of both noise and insufficient knowledge of the function, and only probabilistic safety can be guaranteed when making decisions. The early safe exploration papers proposed their method to guarantee that every decision is safe with high probability, which is also a probabilistic guarantee in the problem they actually addressed. Our problem formulation relaxes this probabilistic threshold, and we expect to achieve a more efficient and practical method.

---

> > > ### Comment · Reviewer_N9rb · 2023-11-18
> > > **Further comments**
> > >
> > > Thanks to the authors for their reply.
> > >
> > > __Optimization over probabilistic safety constraints__
> > >
> > > From the response, it seems like the authors are considering the same problem setting as [1, 2] but substituting $Pr(c(z)) > 1 - \delta$ for the equivalent $Pr(g(z) > h) > \alpha$. Considering this, the problem setting is not novel.
> > >
> > > ##### _In this sense, we __introduce__ a probabilistic version of the safety constraint, which requires that each sample point is safe with a probability above predefined threshold $\alpha$._
> > >
> > > The authors do not _introduce_ the problem setting that is described, so this wording should be changed.
> > >
> > >
> > > __Regularity assumption over latent space.__
> > >
> > > I believe the authors misunderstand my concern here. I am questioning the Assumption itself, namely that the latent space is well-behaved. In HeSBO, the employes subspace is an $\epsilon$-subspace embedding [3] _by construction_ (and not _by assumption_), which in turn enables the construction of a theoretically sound method. Assuming convenient (yet unproven, and arguably unintuitive) properties of the subspace and building theory upon the same convenient properties does not produce sound theoretical contributions.
> > >
> > > As hinted to previously: _If_ the encoding and decoding steps are shown to be $\epsilon$-subspace embeddings, then the theoretical contributions hold merit. Such an embedding is not one to simply be assumed.
> > >
> > > __Assumptions about noise observation.__
> > >
> > > Here, I am encouraging the authors to be consistent throughout the paper. Right now, it is first assumed that observations are noisy, and later assumed to be noiseless.
> > >
> > > __Methodology clarification.__
> > >
> > > Thanks for adding these details. Regarding the UCB definition, I am uncertain why the authors would go with this formulation, as the max over the interval is clearly $\mu(x) \mathbf{+} \beta_t\sigma(x)$. Including the details in the updated version of the paper would, in my view, improve the methodology section.
> > >
> > > __The novelty of the method.__
> > >
> > > The authors say that the proposed method _guarantees_ safety. This mainly hinges on assuming convenient properties, so I am hesitant to agree.
> > >
> > > As a final remark, I would encourage the authors to update the paper with the proposed changes during the rebuttal.
> > >
> > > I am now more confident in my assessment of this paper, and will not vouch for acceptance.
> > >
> > > ### References
> > > [1] Ryan-Rhys Griffiths and José Miguel Hernández-Lobato. Constrained Bayesian optimization for automatic chemical design using variational autoencoders. _Chemical Science_, 2020.
> > >
> > > [2] M. A. Gelbart, J. Snoek and R. P. Adams, Bayesian optimization with unknown constraints. _Proceedings of the
> > > Thirtieth Conference on Uncertainty in Artificial Intelligence_, 2014.
> > >
> > > [3] Sarlos, T. Improved approximation algorithms for large matrices via random projections. _Proceedings of the 47th Annual
> > > IEEE Symposium on Foundations of Computer Science_, 2006.

---

> ### Author Response · Authors · 2023-11-20
>
> Thank you for your quick feedback. We clarify your further comments below.
>
> **Optimization over probabilistic safety constraints.**
>
> We argue that the problem setting in [1, 2] is different from our problem setting. They actually address a constrained optimization problem that aims to find a feasible point that satisfies the probability constraint, neglecting safety during the optimization process. Our problem setting requires that each sample satisfies the probability constraint and considers safety throughout the optimization process.
>
> **Regularity assumption over latent space.**
>
> We think the regularity assumption of the latent space is the prerequisite for using GP to model functions over the latent space. All latent space BO algorithms used GP over the latent space and achieved success on various different high-dimensional problems [3, 4]. Therefore, we believe that it is not a convenient assumption. The regularity assumption over the latent space also does not means the norm-bound is preserved through the decoding step. Since the non-linearity and distortion of the decoder, the function regularity over the original and latent space mat not be able to use the same RKHS to describe. That’s why the $\epsilon$-subspace assumption is introduced to derive our theoretical result.
>
> We also argue that the $\epsilon$-subspace assumption is not a convenient assumption. This assumption can certainly be satisfied by some linear embeddings, such as PCA or random embeddings. In this paper, we use IRVAE to harness the power of deep learning models to learn a better representation while maintaining the approximated isometry. Our further experiment also empirically demonstrates the distance preserving ability of IRVAE as mentioned in the previous rebuttal.
>
> **Assumptions about noise observation.**
>
> Thank you for your suggestions on the consistency of our paper. As mentioned in the previous rebuttal, our theoretical is built over noiseless setting, but the real clinical experiment shows that our proposed method successfully optimizes the general safe optimization problem under noisy observation. We will improve our presentation in the next version to reduce this confusion.
>
> **Methodology clarification.**
>
> Thank you for pointing out the confusion. We will add more details to make the paper clearer.
>
> **The novelty of the method.**
>
> As mentioned above, we argue that the assumptions we use in this paper are not convenient assumptions. The regularity assumptions are the prerequisite of all latent space BO methods, and the $\epsilon$-subspace assumption can be satisfied using some linear projection or approximately satisfied by some distance-preserved autoencoder. We believe our result of safety guarantee is novel.
>
> Our experiment also empirically verifies our theoretical result, where OLLSO consistently outperforms existing constrained BO methods or safe BO methods in terms of both optimization performance and safety guarantee. Our clinical experiment also demonstrates the success of OLLSO in optimizing real-world high-dimensional safety-critical problems. We believe that our practical contribution is novel.
>
> Overall, we believe that our proposed method is novel in both algorithmic and practical aspects, and is both practical and theoretically verifiable to safely and efficiently optimize real-world safety-critical problems.
>
> [1] Ryan-Rhys Griffiths and José Miguel Hernández-Lobato. Constrained Bayesian optimization for automatic chemical design using variational autoencoders. Chemical Science, 2020.
>
> [2] M. A. Gelbart, J. Snoek and R. P. Adams, Bayesian optimization with unknown constraints. Proceedings of the Thirtieth Conference on Uncertainty in Artificial Intelligence, 2014.
>
> [3] Tripp, Austin, Erik Daxberger, and José Miguel Hernández-Lobato. "Sample-efficient optimization in the latent space of deep generative models via weighted retraining." Advances in Neural Information Processing Systems 33 (2020): 11259-11272.
>
> [4] Grosnit, Antoine, et al. "High-dimensional Bayesian optimisation with variational autoencoders and deep metric learning." arXiv preprint arXiv:2106.03609 (2021).

---

### Official Review · Reviewer_WSiY · 2023-11-01

**Soundness:** 2 fair
**Presentation:** 2 fair
**Contribution:** 2 fair
**Rating:** 5
**Confidence:** 3

**Summary:**

The paper proposes a comprehensive solution for efficient, high-dimensional, safe exploration. Essentially, it extends recent advancements in latent space optimization (LSO) and local optimization for global (constrained) optimization (SCBO) into the realm of Bayesian optimization with safety constraints. The paper offers theoretical justification for the proposed VAE-based dimension reduction method in safe exploration when assuming the regularized VAE could achieve $\epsilon$-subspace embedding.

**Strengths:**

1. The paper is well-organized, with sufficient visualization illustrating the key concepts and results. The design of the proposed algorithm is clearly demonstrated. The empirical study shows its improvement over baselines in various metrics.

2. The intermediate evidence substantiates the effectiveness of the proposed IRVAE over traditional VAE in uncertainty quantification in the latent space.

3. The theoretical analysis justifies safe exploration for optimization with Gaussian processes in the latent space.

**Weaknesses:**

1. Though the discussion involves random-projection-based dimension reduction methods, it lacks direct comparison with the proposed IRVAE, especially when the assumption is shared. Specifically, HESBO [1] offers a comprehensive discussion of the feasibility of $\epsilon$-subspace embedding and its impact on downstream uncertainty quantification when applying popular kernels. The recent follow-up work [2] further advances this direction by reducing the risk of losing global optimum in the embedding. A direct comparison could better motivate the proposed IRVAE, which lacks a similar theoretical guarantee and requires additional assumption 3 to proceed with the analysis.

2. The related work part also misses a section dedicated to distance-preserving dimension reduction for uncertainty quantification, which is closely related and should have been compared to the proposed IRVAE. For example, [3] and [4] focus explicitly on distance preserving (feature collapse) in the latent space for uncertainty quantification. Additionally, a similar framework, BALLET [5], relying on superlevel-set identification on top of confidence bounds and deep kernel for high-dimensional Bayesian optimization, is missing in the discussion. There is potential for incorporating its regret analysis to enhance the significance of the proposed paper further.

3. Given existing work in distance preserving dimension reduction, local optimization for efficient global optimization, and safe exploration for optimization with Gaussian processes, the general novelty of the proposed method is very limited.

**References**

[1] Nayebi, Amin, Alexander Munteanu, and Matthias Poloczek. "A framework for Bayesian optimization in embedded subspaces." In International Conference on Machine Learning, pp. 4752-4761. PMLR, 2019.

[2] Papenmeier, Leonard, Luigi Nardi, and Matthias Poloczek. "Increasing the scope as you learn: Adaptive Bayesian optimization in nested subspaces." Advances in Neural Information Processing Systems 35 (2022): 11586-11601.

[3] Ober, Sebastian W., Carl E. Rasmussen, and Mark van der Wilk. "The promises and pitfalls of deep kernel learning." In Uncertainty in Artificial Intelligence, pp. 1206-1216. PMLR, 2021.

[4] van Amersfoort, Joost, Lewis Smith, Andrew Jesson, Oscar Key, and Yarin Gal. "On feature collapse and deep kernel learning for single forward pass uncertainty." arXiv preprint arXiv:2102.11409 (2021).

[5] Fengxue Zhang, Jialin Song, James Bowden, Alexander Ladd, Yisong Yue, Thomas A. Desautels, and Yuxin Chen. 2023. Learning regions of interest for Bayesian optimization with adaptive level-set estimation. In Proceedings of the 40th International Conference on Machine Learning (ICML'23), Vol. 202. JMLR.org, Article 1745, 41579–41595.

**Questions:**

Could the author further clarify the results shown in Figure 3? What are the legends in (b) stand for? Does the bar plots in (c) show the cumulative results after 500 evaluation corresponding to the results in (b)?

---

> ### Author Response · Authors · 2023-11-17
> **Rebuttal [1/2]**
>
> **Comparison with random-embedding BO**
>
> Thanks for your suggestions on adding random-embedding-based methods for comparison. We follow your suggestion and run HesBO and BAxUS on a musculoskeletal model control task and a neural stimulation task with no safety constraints in simulation. Due to the algorithmic mechanism of HesBO and BAxUS, we cannot directly use the same initial point as OLLSO. Therefore we randomly sample initial points from their corresponding latent space. In HesBO, we set the same latent dimension number as in OLLSO. The following tables are the best objective function values found by algorithms (shown as mean $\pm$ 1 std).
>
> | Algorithm | Muscle                         | SCS-IL                     | SCS-RF                     | SCS-TA                     |
> |-----------|--------------------------------|----------------------------|----------------------------|----------------------------|
> | OLLSO     | $\boldsymbol{284.08\pm305.95}$ | $\boldsymbol{0.39\pm0.03}$ | $\boldsymbol{0.37\pm0.00}$ | $\boldsymbol{0.38\pm0.02}$ |   |
> | HesBO     | $-747.8\pm124.53$              | $0.34\pm0.03$              | $0.30\pm0.04$              | $0.36\pm0.05$              |
> | BAxUS     | $-619.36\pm302.83$             | $0.36\pm0.04$              | $0.35\pm0.02$              | $0.34\pm0.05$              |
>
> | Algorithm | SCS-BF                     | SCS-ST                     | SCS-GA                     |
> |-----------|----------------------------|----------------------------|----------------------------|
> | OLLSO     | $\boldsymbol{0.28\pm0.03}$ | $\boldsymbol{0.26\pm0.02}$ | $\boldsymbol{0.22\pm0.01}$ |
> | HesBO     | $0.25\pm0.01$              | $0.23\pm0.0$               | $0.19\pm0.02$              |
> | BAxUS     | $0.27\pm0.02$              | $0.24\pm0.03$              | $0.19\pm0.05$              |
>
> We observe OLLSO still outperforms HesBO and BAxUS across all tasks, even when optimizing under safety constraint. We think using IRVAE enables utilizing the pre-collected unlabelled data to learn a better representation than random projection.
>
> **Missing discussion about related works**
>
> Thank you for pointing out the missing discussion of distance-preserving dimension reduction for uncertainty quantification. The listed papers are relevant to our work, and we will include them in the discussion part in the next version of our paper. BALLET is a good work incorporating deep kernel learning with level set estimation to tackle high-dimensional problems, and we will try to get insights from this work and conduct regret analysis in the future.
>
> **Experiment result clarification**
>
> 1. Figure 3 (b) shows the clinical result in neural stimulation therapy optimization, where IL (iliopsoas), RF (rectus femoris), TA (tibialis anterior), BF (biceps femoris) are different groups of target muscles on the lower limb, which are the main muscle groups that control motor funuction. The "L" (Left) or "R" (Right) after the underline are the sides of human lower limbs.
>
> 2. Figure 3 (c) shows the simulation result using a computational spinal cord model, where the experiment result is detailed in Section 6.3.1. Bar plot in Figure 3 (c) shows the best feasible objective function value (Objective, top), safe decision ratio of all samples (Safe \%, middle), and cumulative safety violation (Violation, bottom) after 800 samples.
>
> Thank you for pointing out the confusion part, and we will improve our experiment result presentation in the next version.

---

> ### Author Response · Authors · 2023-11-17
> **Rebuttal [2/2]**
>
> **The novelty of our work.**
>
> In this paper, we propose a practical method to achieve safe and efficient optimization of high-dimensional functions. To our best knowledge, no existing work focuses on guaranteeing safety during the high-dimensional optimization.
>
> We present the novelty of OLLSO in both algorithmic and practical aspects.
>
> In terms of latent space optimization, previous latent BO works neglect the geometry-preserving during unsupervised representation learning, which is crucial to extend safety guarantee from the latent space to the original space. Many of them achieve latent space shaping using collected labeled data [1, 2]. We are the first to use distance-preserved VAE to address safety issue in latent space optimization, and derive the theoretical probabilistic safety guarantee of our proposed method.
>
> The local region adaptation of OLLSO is also different from existing
> trust-region based BO methods [3, 4] which neglect the safety during the optimization procedure. They would also discard all previous sample points when restarting the trust region. OLLSO updates the trust region in a more safety-sensitive way, and keeps all previous samples when restarting the trust region, ensuring a safer optimization procedure with theoretical guarantee.
>
> In terms of safe optimization, previous GP-based safe optimization method are limited to low-dimensional problems (typically below 10, [5, 6]). OLLSO addresses this issue by introducing local search mechanism upon optimistic safety region identification, and is able to handle problem with hundreds of dimensions.
>
> Our experimental result also shows that OLLSO is not a direct application to comprise exsiting algorithms. Figure 2 and Figure 3 show that simply combining IRVAE with existing safe BO (CONFIG (L)) or trust region-based constrained BO (SCBO (L)) perform worse than OLLSO in terms of both optimization performance and safety guarantee. Our simulation results shows that OLLSO significantly outperforms existing safe BO methods, constrained BO algorithms, and direct combinations with dimension reduction models.
>
> We want to emphasize that OLLSO is a PRACTICAL method that can safely optimize high-dimensional real-world problems. In addition to state-of-the-art simulation performance, our real clinical experiments show that OLLSO successfully optimizes neural stimulation for the control of human muscles. While the majority of our cited papers conduct their experiments only in simulation or synthetic problems, OLLSO is both practical and theoretically verifiable to safely and efficiently optimize real-world safety-critical problems.
>
> [1] Tripp, Austin, Erik Daxberger, and José Miguel Hernández-Lobato. "Sample-efficient optimization in the latent space of deep generative models via weighted retraining." Advances in Neural Information Processing Systems 33 (2020): 11259-11272.
>
> [2] Grosnit, Antoine, et al. "High-dimensional Bayesian optimisation with variational autoencoders and deep metric learning." arXiv preprint arXiv:2106.03609 (2021).
>
> [3] Eriksson, David, et al. "Scalable global optimization via local Bayesian optimization." Advances in neural information processing systems 32 (2019).
>
> [4] Eriksson, David, and Matthias Poloczek. "Scalable constrained Bayesian optimization." International Conference on Artificial Intelligence and Statistics. PMLR, 2021.
>
> [5] Sui, Yanan, et al. "Safe exploration for optimization with Gaussian processes." International conference on machine learning. PMLR, 2015.
>
> [6] Xu, Wenjie, et al. "Constrained efficient global optimization of expensive black-box functions." International Conference on Machine Learning. PMLR, 2023.

---

> ### Author Response · Authors · 2023-11-21
>
> Dear Reviewer WSiY,
>
> Thank you again for your valuable feedback. We have posted the additional experiment comparing with random-projection-based BO algorithms and the clarification about the novelty of our work. As the deadline is less than two days away, we look forward to your further feedback to make sure that all your concerns are addressed.
>
> Best,
>
> OLLSO authors

---

> > ### Comment · Reviewer_WSiY · 2023-11-23
> >
> > Dear Authors,
> >
> > I appreciate the response to my comments and questions. I agree with my fellow reviewers that the paper's novelty is one of the most outstanding concerns. The author's discussion over the connection to the existing work actually confirms the concern. Combining existing ideas in a principled way could still be significant, either with additional empirical insights or a rigorous theoretical justification for the integration. Through the rebuttal, some progressions are made and should be added to the revised version. Yet, I believe the added results and discussions are insufficient. Hence, I intend not to merit the current presentation.
> >
> > Thanks.

---

> > > ### Author Response · Authors · 2023-11-23
> > >
> > > Thank you for your further comment. We will improve our presentation in the revised version according to your valuable feedback. We believe that our method is constructed in a principled way, where each component is introduced to address a specific challenge in high-dimensional safe optimization. Our theoretical results verify the safe guarantee of OLLSO and our experimental results demonstrate the empirical insights of both the efficiency and safety of OLLSO in high-dimensional safe-critical problems.

---

### Official Review · Reviewer_Cfdg · 2023-11-03

**Soundness:** 2 fair
**Presentation:** 3 good
**Contribution:** 2 fair
**Rating:** 5
**Confidence:** 3

**Summary:**

This paper develops a solution approach to the safe BO task on high-dimensional space. The key idea is to find a distance-preserving low-dimensional embedding on the original input space. This is achieved by adopting a previously developed Isometrically Regularized VAE (IRVAE). Once the IRVAE has been built, a previously established safe exploration variant of GP-UCB is applied.

Under certain assumptions, a certain probability of meeting the safety requirement can be theoretically guaranteed. The proposed method is applied on a variety of real-world, practical experiment. The key contribution here are the empirical studies on real-world dataset and the theoretical analysis.

**Strengths:**

The paper aims to address a very important problem. Existing literature has also been sufficiently reviewed.
Most importantly, I really appreciate that the paper features a pretty interesting set of very real-world, practical experiments.
There is also a result that helps translate the safety guarantee from the latent space to the original space.

**Weaknesses:**

Overall, I appreciate the strong empirical studies of this paper. All experiments are based on very real-world application & that is great.

However, I am also concerned that the algorithmic contribution of this paper is too incremental: if I understand this paper correctly, it comprises two separate phases (learning low-dim embedding & doing safe BO on the low-dim space) and each phase is a direct application of an existing algorithm.

In addition, I am also not sure what the current theoretical analysis implies. I understand that Theorem 1 is established to translate a probabilistic constraint on the latent space to another on the original space.

It seems the math suggest a translation on the UCB of the safety function prediction while in practice, we would want to establish a probabilistic bound on a user-specified constraint g(x) >= h where h is given. Theorem 1 does not provide any handle on this.

---

Minor point:

The algorithmic exposition is also unclear at several points. For example, in Algorithm 1 (line 6), it is not clear how L_t is defined. Furthermore, how do we update it in line 6?

**Questions:**

Based on the above, I have two specific questions:

1. Could you flesh out Algorithm 1 mathematically in the rebuttal?
2. Could you elaborate on how Theorem 1 can be positioned to guarantee that with a certain algorithmic configuration, the proposed algorithm would induce P(g(x) >= h) >= alpha for a given h?
3. Based on 1. and 2., it would be good to highlight the non-triviality of putting together the ideas of safe exploration & BO on latent space. Otherwise, a simple loose coupling of these ideas is somewhat below bar for me -- I of course appreciate the practical empirical experiment -- I think the set of experiments is good but that alone is probably not enough to meet the acceptance bar.

---

> ### Author Response · Authors · 2023-11-17
> **Rebuttal [1/2]**
>
> Thank you for your appreciation of our paper and your extensive comments. We clarify our work based on the weaknesses and questions you mentioned.
>
> **Algorithm 1 explained in more mathematical details.**
>
> * (Line 1) we train an IRVAE using the unlabelled dataset $D_u^{\mathcal{X}}$ by adding an isometric regularization loss term over the original loss function (Line 1). Our training objective is
>
>     $\min_{\theta} \mathcal{L}(\theta) + \mathcal{F}(\theta)$,
>
> where $\theta$ is the parameters of the IRVAE, $\mathcal{L}(\theta)$ is the original loss function (typically using evidence lower bound in VAE), and $\mathcal{F}(\theta)$ is the additional regularization term proposed by [1] to shape the learned representation towards isometric.
>
> * (Line 2-3) We initialize the sample trajectory as an empty set, and start optimization loop. The sample trajectory $\zeta_{t}$ is defined as a set of tuples {$(\boldsymbol{x}_t, y_t^f, y_t^g)$}.
>
> * (Line 4-5) At each optimization round $t$, we use the trained IRVAE to map original sampled data to the latent space, and update the objective and safety GP model using Eq.2.
>
> * (Line 6) We update the local region $L_t$ based on the sample trajectory $\zeta_{t-1}$. $L_t$ is a trust region defined by a center (using current best feasible point) and a length $l$.  We denote successful/failure of the optimization round using the last sample result: A sampling round is considered "successful" if it finds a better reward while maintaining comprehensive safety. Conversely, it is labelled a "failure" if any unsafe points are found or if there is no discernible improvement. The side length is adjusted—increased ($l_t = 2l_{t-1}$) for successes and decreased ($l_t = 0.5l_{t-1}$) for failures—upon reaching a preset threshold.
>
> * (Line 7-8) We use UCB of the safety GP to identify the safe region, and maximize the acquisition function (using TS here) to get next latent sample.
>
> * (Line 9-12) We project the latent sample back to original space, evaluate the function values and update the sampled dataset and sample trajectory.
>
> We will improve our presentation to help readers better understand our proposed method.
>
> **Link the theoretical result with the optimization objective.**
>
> In Theorem 1, we prove that with proper choice of the confidence bound scalar $\beta$, the selected safety value is lager than the estimated upper confidence bound with a probability larger than $\alpha$ (that is, $Pr(g(\boldsymbol{x}_t) \geq u(\boldsymbol{x}_t))\geq\alpha$). During optimization, we only consider points whose UCB is larger than the safety threshold $h$ as candidates (Line 7 in Algorithm 1). In this way we can guarantee that the selected safety value is lager than the safety threshold $h$ with a probability larger than $\alpha$ (that is, $Pr(g(\boldsymbol{x}_t) \geq h)\geq Pr(g(\boldsymbol{x}_t) \geq u(\boldsymbol{x}_t)) \geq \alpha)$.

---

> ### Author Response · Authors · 2023-11-17
> **Rebuttal [2/2]**
>
> **The novelty of our work.**
>
> In this paper, we propose a practical method to achieve safe and efficient optimization of high-dimensional functions. To our best knowledge, no existing work focuses on guaranteeing safety during the high-dimensional optimization.
>
> We present the novelty of OLLSO in both algorithmic and practical aspects.
>
> OLLSO is a novel safe optimization algorithm. Previous GP-based safe optimization method are limited to low-dimensional problems (typically below 10, [2, 3]). OLLSO addresses this issue by introducing local search mechanism upon optimistic safety region identification. It is able to handle problem with hundreds of dimensions.
> The local region adaptation of OLLSO is also different from existing
> trust-region based BO methods[4, 5] which neglect the safety during the optimization procedure. Those methods would also discard all previous sample points when restarting the trust region. OLLSO updates the trust region in a more safety-sensitive way, and keeps all previous samples when restarting the trust region, ensuring a safer optimization procedure with theoretical guarantee.
>
> OLLSO has a dimension reduction component to handle very high-dimensional or discrete input spaces. Previous latent BO works neglect the geometry-preserving during unsupervised representation learning, which is crucial to extend safety guarantee from the latent space to the original space. Many of them achieve latent space shaping using collected labeled data[6, 7]. We are the first to use distance-preserved VAE to address safety issue in latent space optimization, and derive the theoretical probabilistic safety guarantee of our proposed method.
>
> Our experimental result also explains that OLLSO is not a direct application to comprise existing algorithms. Figure 2 and Figure 3 show that directly combining IRVAE with existing safe BO (CONFIG (L)) or trust region-based constrained BO (SCBO (L)) perform worse than OLLSO in terms of both optimization performance and safety guarantee. Our simulation results show that OLLSO significantly outperforms existing safe BO methods, constrained BO algorithms, and direct combinations with dimension reduction models.
>
> We want to emphasize that OLLSO is a PRACTICAL method that can safely optimize high-dimensional real-world problems. In addition to state-of-the-art simulation performance, our real clinical experiments show that OLLSO successfully optimizes neural stimulation for the control of human muscles. While the majority of our cited papers conduct their experiments only in simulation or synthetic problems, OLLSO is both practical and theoretically verifiable to safely and efficiently optimize real-world safety-critical problems.
>
> [1] Yonghyeon, L. E. E., et al. "Regularized autoencoders for isometric representation learning." International Conference on Learning Representations. 2021.
>
> [2] Sui, Yanan, et al. "Safe exploration for optimization with Gaussian processes." International conference on machine learning. PMLR, 2015.
>
> [3] Xu, Wenjie, et al. "Constrained efficient global optimization of expensive black-box functions." International Conference on Machine Learning. PMLR, 2023.
>
> [4] Eriksson, David, et al. "Scalable global optimization via local Bayesian optimization." Advances in neural information processing systems 32 (2019).
>
> [5] Eriksson, David, and Matthias Poloczek. "Scalable constrained Bayesian optimization." International Conference on Artificial Intelligence and Statistics. PMLR, 2021.
>
> [6] Tripp, Austin, Erik Daxberger, and José Miguel Hernández-Lobato. "Sample-efficient optimization in the latent space of deep generative models via weighted retraining." Advances in Neural Information Processing Systems 33 (2020): 11259-11272.
>
> [7] Grosnit, Antoine, et al. "High-dimensional Bayesian optimisation with variational autoencoders and deep metric learning." arXiv preprint arXiv:2106.03609 (2021).

---

> > ### Comment · Reviewer_Cfdg · 2023-11-19
> > **Follow-up**
> >
> > Thank you for the detailed response. Your response has addressed my concern about how Theorem 1 provides a handle on the safety guarantee. The extra detail regarding the Algorithm is also helpful.
> >
> > However, your response on the novelty of the paper does not seem to add much more to my original assessment. I already agree that there is a practical contribution here but I maintain that that alone is marginally below bar for acceptance.
> >
> > In particular, the weakness that I highlight is that the algorithm is a direct coupling between GP-UCB and IRVAE, and your response to that is
> >
> > "Our experimental result also explains that OLLSO is not a direct application to comprise existing algorithms. Figure 2 and Figure 3 show that directly combining IRVAE with existing safe BO (CONFIG (L)) or trust region-based constrained BO (SCBO (L)) perform worse than OLLSO in terms of both optimization performance and safety guarantee. Our simulation results show that OLLSO significantly outperforms existing safe BO methods, constrained BO algorithms, and direct combinations with dimension reduction models."
> >
> > Maybe there is still something I have missed here, but this response does not seem to tell me what is there between a vanilla GP-UCB and a vanilla IRVAE. The only (minor) thing that I noticed is the restriction to all point with the UCB exceeding the given safety threshold.
> >
> > --
> >
> > Furthermore, while there is a probabilistic guarantee, it depends on a strong assumption that IRVAE would produce something that fits what is stated in Assumption 1 so overall, it feels like a superficial thing in the sense that we see from the experiment that the method performs well but we still do not know why. Surely, with such an assumption that might not hold, we cannot simply associate the perceived improvement with the current theory.
> >
> > This is the weak point of this paper, which cannot be mitigated with a vanilla coupling of IRVAE and GP-UCB. Perhaps a less trivial integration would be to construct an embedding method such that the key assumption is more likely to hold? I believe such an approach would tie up this loose knot.
> >
> > --
> >
> > Overall, while I agree that the approach has merit, its contribution is marginally below bar because it still has a loose end. Strictly speaking, safety guarantee cannot and should not depend on something that might or might not hold.

---

> > > ### Author Response · Authors · 2023-11-20
> > >
> > > Thank you for your quick feedback! We clarify your concerns below.
> > >
> > > **The novelty of our work.**
> > >
> > > We argue that the algorithm is not a direct coupling between GP-UCB and IRVAE. GP-UCB uses the estimated upper confidence bound of the objective function as the acquisition function to balance exploration and exploitation. Here, we use the upper confidence bound of the safety function to optimistically identify the safe region. We then optimize the acquisition function (Thompson sampling used in the paper) within the safe region to obtain the next sample using the objective GP. Direct coupling between GP-UCB and IRVAE cannot achieve the safe optimization that OLLSO addresses because it only considers objective optimization and neglects safety.
> > >
> > > **The assumptions used in our work.**
> > >
> > > Our theoretical result is based on the regularity assumption (Assumption 1) over the latent space and the assumptions about the distance preserving ability (Definition 1) of the embedding, and we argue that the assumptions used are not trivial.
> > >
> > > We believe that the regularity assumption of the latent space is a prerequisite for using GP to model functions over the latent space. All latent space BO algorithms used GP over the latent space and achieved success on various high-dimensional problems [1, 2]. Therefore, we believe that it is not a convenient assumption.
> > >
> > > We also argue that the $\epsilon$-subspace assumption is not a convenient assumption. This assumption can certainly be satisfied by some linear embeddings, such as PCA or random embeddings. In this paper, we use IRVAE to harness the power of deep learning models to learn a better representation while maintaining the approximated isometry. Our further experiment also empirically demonstrates the distance-preserving ability of IRVAE. In Figure 4, we show that the GP estimation difference between the original space and the latent space of IRVAE is significantly smaller than the latent space of standard VAE. We also compute the pointwise distance of 10,000 randomly sampled points. The results show a high linear correlation of the point-wise distance between the original space and the latent space (R^2 is 0.851 for the musculoskeletal model control task and 0.973 for the neural stimulation task), indicating a good distance preserving ability of IRVAE.
> > >
> > > Overall, we believe that our proposed method is novel in both algorithmic and practical aspects, and is both practical and theoretically verifiable to safely and efficiently optimize real-world safety-critical problems.
> > >
> > > [1] Tripp, Austin, Erik Daxberger, and José Miguel Hernández-Lobato. "Sample-efficient optimization in the latent space of deep generative models via weighted retraining." Advances in Neural Information Processing Systems 33 (2020): 11259-11272.
> > >
> > > [2] Grosnit, Antoine, et al. "High-dimensional Bayesian optimisation with variational autoencoders and deep metric learning." arXiv preprint arXiv:2106.03609 (2021).

---

### Meta-Review · Area_Chair_pUKV · 2023-12-21

**Metareview:**

This work presents a method for safe Bayesian optimization in high dimensions, where each iteration must achieve some constraint threshold with probability $alpha$.  It utilizes a pre-trained isometrically regularized VAE and performs trust region BO within the embedded space. The paper includes very interesting and novel high-dimensional medical applications of Bayesian optimization. However, the principal contribution lies in the safety guarantees of the algorithms. While the experiments in Fig4 do indeed show that IRVAE does indeed preserve distances better than vanilla VAEs, reviewers did not find the experiments to sufficiently demonstrate that IRVAE or the algorithm satisfy the $\epsilon$-subspace assumption.  Ultimately, no reviewer was willing to champion the paper.  This was a very interesting paper to read, there are no compelling high-dimensional safe BO methods, so I encourage the authors to continue to improve the work and resubmit elsewhere.

**Justification For Why Not Higher Score:**

No reviewer was championing paper.

**Justification For Why Not Lower Score:**

N/A

---

### Decision · Program_Chairs · 2024-01-16

Reject